# A host-directed oxadiazole compound potentiates antituberculosis treatment via zinc poisoning in human macrophages and in a mouse model of infection

Alexandra Maure[1], Emeline Lawarée[1], Francesco Fiorentino[2], Alexandre Pawlik[1], Saideep Gona[3], Alexandre Giraud-Gatineau[4], Matthew J. G. Eldridge[5], Anne Danckaert[6], David Hardy[7], Wafa Frigui[1], Camille Keck[1], Claude Gutierrez[8], Olivier Neyrolles[8], Nathalie Aulner[6], Antonello Mai[2,9], Mélanie Hamon[5], Luis B. Barreiro[3], Priscille Brodin[10], Roland Brosch[1], Dante Rotili[2], Ludovic Tailleux[1] *

1 Institut Pasteur, Université Paris Cité, CNRS UMR 6047, Unit for Integrated Mycobacterial Pathogenomics, Paris, France, 2 Department of Drug Chemistry and Technologies, Sapienza University of Rome, Rome, Italy, 3 Department of Genetic Medicine, University of Chicago, Chicago, Illinois, United States of America, 4 Institut Pasteur, Université Paris Cité, CNRS UMR 6047, Biology of Spirochetes Unit, Paris, France, 5 Institut Pasteur, Université Paris Cité, Chromatine et Infection unit, Paris, France, 6 Institut Pasteur, Université Paris Cité, UTechS BioImaging-C2RT, Paris, France, 7 Institut Pasteur, Université Paris Cité, Histopathology Platform, Paris, France, 8 Institut de Pharmacologie et de Biologie Structurale, IPBS, Université de Toulouse, CNRS, UPS, Toulouse, France, 9 Pasteur Institute, Cenci-bolognetti Foundation, Sapienza University of Rome, Rome, Italy, 10 Université de Lille, CNRS, INSERM, CHU Lille, Institut Pasteur de Lille, U1019 - UMR 9017 - CIIL - Center for Infection and Immunity of Lille, Lille, France

* ludovic.tailleux@pasteur.fr

**Data Availability Statement:** All relevant data are within the paper and its Supporting information files. The RNA-Seq fastq files have been deposited

## Abstract

Antituberculosis drugs, mostly developed over 60 years ago, combined with a poorly effective vaccine, have failed to eradicate tuberculosis. More worryingly, multiresistant strains of *Mycobacterium tuberculosis* (MTB) are constantly emerging. Innovative strategies are thus urgently needed to improve tuberculosis treatment. Recently, host-directed therapy has emerged as a promising strategy to be used in adjunct with existing or future antibiotics, by improving innate immunity or limiting immunopathology. Here, using high-content imaging, we identified novel 1,2,4-oxadiazole-based compounds, which allow human macrophages to control MTB replication. Genome-wide gene expression analysis revealed that these molecules induced zinc remobilization inside cells, resulting in bacterial zinc intoxication. More importantly, we also demonstrated that, upon treatment with these novel compounds, MTB became even more sensitive to antituberculosis drugs, in vitro and in vivo, in a mouse model of tuberculosis. Manipulation of heavy metal homeostasis holds thus great promise to be exploited to develop host-directed therapeutic interventions.

## Introduction

Antibiotics have considerably extended human's life span by changing the outcome of bacterial infections. With this medical breakthrough came the emergence of antimicrobial

in NCBI's Gene Expression Omnibus and are accessible through GEO Series accession number GSE222412.

**Funding:** This research project was funded by the Institut Pasteur (grant to RB), the Georges, Jacques and Elias Canetti Award (grant to LT) and the French National Research Agency (grant ANR Mustart 20-PAMR-0005 to RB and LT). We gratefully acknowledge the UTechS Cytometry and Biomarkers and the UTechS Photonic BioImaging (PBI) /C2RT of Institut Pasteur (Paris, France). PBI, part of the France–BioImaging infrastructure network (ANR-10–INSB–04; Investments for the Future) kindly acknowledge the support of the Région Île-de-France (DIM1 Health) for the use of the Opera Phenix. AM was supported by the Fondation pour la Recherche Médicale (FRM FDM201806006250). DR acknowledges Sapienza Ateneo Project 2021 RM12117A61C811CE and Regione Lazio PROGETTI DI GRUPPI DI RICERCA 2020, project ID: A0375-2020-36597. AnM acknowledges the project FISR2019_00374 MeDyCa. The funders had no role in study design, data collection and analysis, decision to publish, or preparation of the manuscript.

**Competing interests:** The authors have declared that no competing interests exist.

**Abbreviations:** 3-MA, 3-methyladenine; ADC, albumin dextrose catalase; ALT, alanine aminotransferase; ANOVA, analysis of variance; AST, aspartate aminotransferase; Atc, anhydrotetracycline; BAF, Bafilomycin A1; BDQ, bedaquiline; BHI, brain–heart infusion; CFU, colony-forming unit; CQ, chloroquine; DRC, dose–response curve; EDTA, ethylenediaminetetraacetic acid; FBS, fetal bovine serum; GFP, green fluorescent protein; GM-CSF, granulocyte-macrophage colony-stimulating factor; GO, gene ontology; GSH, glutathione; HDT, host-directed therapy; HE, hematoxylin–eosin; HeLa, Henrietta Lacks; IFN, interferon; INH, isoniazid; LB, Luria Bertani; LC3B, light-chain 3B; M-CSF, macrophage colony-stimulating factor; MDR, multidrug resistant; MEF, mouse embryonic fibroblast; MIC, minimal inhibitory concentration; MOI, multiplicity of infection; MTB, *Mycobacterium tuberculosis*; mTOR, mechanistic target of rapamycin; NAC, N-Acetyl-L-cysteine; RIF, rifampicin; ROS, reactive oxygen species; RT, room temperature; RT-qPCR, quantitative reverse transcription PCR; SIRT2, sirtuin 2; TB, tuberculosis; V-ATPase, vacuolar-type H+-ATPase; WT, wild-type.

resistance worldwide, representing one of the biggest threats to human health. Tuberculosis (TB) is no exception, with the clock ticking for TB care and prevention. TB is a disease caused by the bacterium *Mycobacterium tuberculosis* (MTB), which is spread from person to person through the air. Although significant progress has been made to tackle TB over the last few decades, with the discovery of major antituberculosis drugs [1], the disease still kills over 1.6 million people annually [2]. Worryingly, multidrug resistant (MDR) strains of MTB, which are resistant to both the frontline anti-TB drugs rifampicin (RIF) and isoniazid (INH), have emerged and are widespread worldwide. In 2022, 410,000 patients were diagnosed with RIF-resistant TB or MDR-TB [2]; 20% of those patients also show resistance to fluoroquinolones, an important drug class for the treatment of drug-resistant TB [2]. Typically, drug-sensitive TB can be cured by a 6-month treatment, combining up to 4 antibiotics, namely, INH, RIF, ethambutol, and pyrazinamide. Curing MDR-TB is more difficult, despite recent advances [3]. While new antibiotics are being developed and brought to the clinic, resistance to these molecules is rapidly detected. New strategies are thus urgently needed to prevent the emergence of drug resistance and to shorten treatment duration.

Host-directed approaches are a promising strategy to be used in adjunct with existing or future antibiotics [4,5]. Several host-directed therapies (HDTs) to treat TB have been developed and mostly act in (i) the manipulation of host antimycobacterial pathways that are blocked or altered by MTB to promote its survival; (ii) the potentiation of antimicrobial host immune defense mechanisms; or (iii) the amelioration of immunopathology [5]. Macrophages represent a prime target for HDT as they play a key role in the outcome of a mycobacterial infection. On one hand, they orchestrate the formation of granulomas, present mycobacterial antigens to T cells and can kill the bacillus upon IFN-γ activation [6]. On the other hand, macrophages are the primary host niche for MTB, which has developed different strategies to survive and multiply inside the macrophages' phagosome. These include prevention of phagosome acidification [7], inhibition of phagolysosomal fusion [8], and phagosomal rupture [9,10].

Some in vitro studies have demonstrated that HDT-compounds are indeed effective at increasing macrophage resistance to MTB infection [4]. These molecules include repurposed FDA-approved drugs or new compounds that have been identified by high-throughput screening of MTB-infected cells. Most of them promote phagosome maturation (for instance, tyrosine kinase inhibitors), activate autophagy (such as inhibitors of the mechanistic target of rapamycin (mTOR)), induce antimicrobial peptides, or inhibit lipid body formation [4]. Here, we identified new molecules able to limit MTB growth in human macrophages by inducing zinc remobilization inside cells, resulting in bacterial poisoning. Interestingly, these compounds also potentiate the activity of known anti-TB drugs not only in vitro but also in vivo in a TB mouse model.

## Results

### Identification of a compound that inhibits the intracellular MTB growth

We and others have previously shown that numerous epigenetic modifications occur in myeloid cells upon MTB infection [11–13]. These changes are either part of the initiation of the host response or an immune escape mechanism from MTB. MTB indeed manipulates epigenetic host-signaling pathways to subvert host immunity [11]. Reprogramming the host immune system by a compound targeting the host epigenome may thus lead to a better control of the bacterial infection [14]. To tackle this hypothesis, we screened an in-house library of 157 epigenetics-related compounds in MTB-infected macrophages using high-content imaging. This library comprises activators and inhibitors of enzymes that carry out various epigenetic

modifications. For instance, it includes molecules that modulate the activity of DNA methyltransferases, histone methyltransferases and demethylases, histone acetyltransferases and deacetylases, and poly (ADP-ribose) polymerases (S1 Table).

Briefly, human monocyte-derived macrophages were infected with MTB expressing green fluorescent protein (GFP) at a multiplicity of infection (MOI) of 0.5, before being seeded in a 384-well plate; each well contained a different compound at a final concentration of 10 μM. RIF was used as a positive control. After 96 h of infection, cells were fixed and nuclei were stained with Hoechst 33342. Fluorescence was analyzed by automated confocal microscopy. Cell toxicity was determined by comparing the number of nuclei stained by Hoechst 33342 in treated wells to the negative control well (0.1% DMSO). No macrophage toxicity or any adverse effects of DMSO on the growth of MTB were observed at this concentration of DMSO (S1A and S1B Fig). The quality of our assay was determined by calculating the Z'-factor (15). The Z'-factor was superior to 0.5 indicating the robustness of our high-content screening (S1C Fig). Only compounds associated with a cell viability greater than 75% were considered for further analysis. A total of 106 molecules were nontoxic to human macrophages (Fig 1A). Inhibition of intracellular MTB growth in presence of the remaining compounds was then assessed by determining the GFP area per cell, which is correlated to the amount of intracellular MTB (S1D and S1E Fig). We identified 1 molecule, MC3465, which reduces the bacterial growth by 55% compared to the control (Fig 1A and 1B). To validate MC3465 as a promising hit, we repeated the same experiment with 8 replicates of DMSO, RIF, and MC3465. We then calculated the z-score. The z-score was less than −5 (S1F Fig), indicating that MC3465 consistently and significantly affected intracellular MTB growth. This result was confirmed by enumerating the number of bacteria inside macrophages treated with MC3465 for 24 h and 96 h. MC3465 appears to be bacteriostatic or at least significantly reduces the growth of MTB. A 2.25- and 4-fold reduction in colony-forming units (CFUs) was observed after 24 and 96 h of treatment, respectively, compared to the DMSO control (Fig 1C). There are substantial differences in the virulence of MTB strains. These differences could lead to an attenuated activity of our compound. We thus tested the efficacy of MC3465 in macrophages infected with highly virulent clinical isolates of MTB, namely, CDC1551, GC1237, and Myc5750 (the last 2 belonging to the Beijing family) [16–18]. After 96 h treatment, we observed a substantial decrease in bacterial survival rates with MC3465 (Fig 1D). This compound did not show any cell toxicity, even at higher concentration (50 μM) over an incubation period of 7 d (S2A Fig). MC3465 was equally active in macrophages differentiated in the presence of granulocyte-macrophage colony-stimulating factor (GM-CSF) or macrophage colony-stimulating factor (M-CSF), 2 cytokines known to prime toward pro-inflammatory (M1) and alternatively activated (M2) phenotypes [19] (S2B Fig).

We next tested whether MC3465 also inhibited the intracellular bacterial growth of other bacteria. Macrophages were infected with *Listeria monocytogenes*, *Salmonella enterica serovar* Typhimurium, and *Escherichia coli* for 24 h, in presence of MC3465 (10 μM) or with DMSO alone. Cells were then lysed and CFUs were enumerated. Our results show that MC3465 does not impede the intracellular growth of *L. monocytogenes*, *S.* Typhimurium, and *E. coli*, indicating that the action of this compound could be specific to MTB-infected cells (Figs 1E and S2C).

MC3465 is supposed to act as an inhibitor of the human NAD$^+$-dependent deacetylase sirtuin 2 (SIRT2) [20]. However, it could have a direct inhibitory effect on the bacteria. To verify the host-directed effect of this molecule, we monitored the MTB growth in liquid medium by measuring the optical density (OD$_{600}$ nm) or by counting the number of CFUs at different time points. Even at a higher concentration (50 μM), MC3465 does not affect MTB growth (Figs 1F and S2D). To further confirm that MC3465 targets only intracellular MTB, bacteria

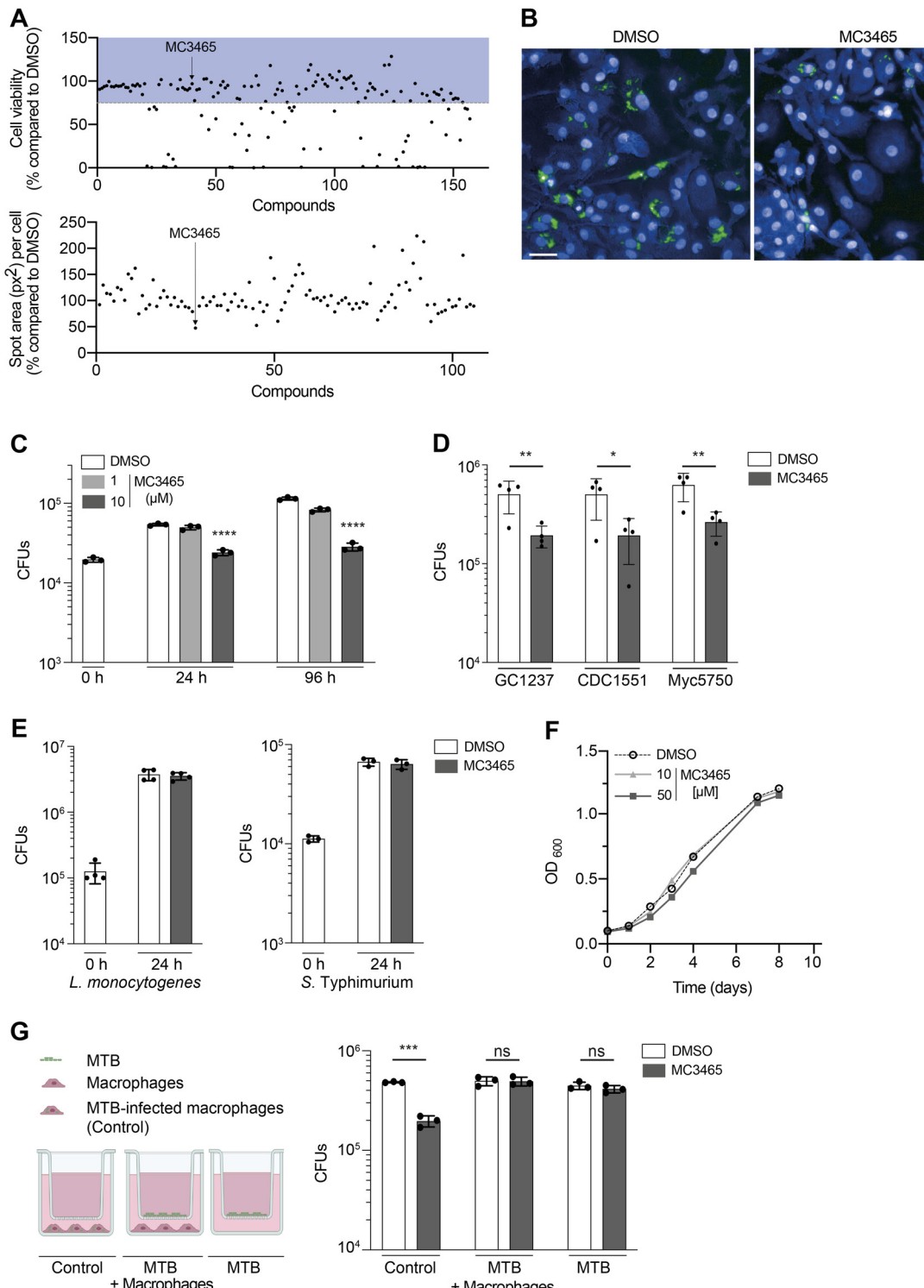

**Fig 1. Identification of new compounds inhibiting the intracellular growth of MTB. (A)** Human monocyte-derived macrophages were infected with GFP-expressing MTB (GFP-MTB) and incubated with epigenetics-related compounds. After 96-h treatment, cells were labeled with Hoechst 33342 and HCS CellMask Blue. Fluorescence was analyzed by the Opera Phenix Plus High-Content Screening System. Quantification of GFP staining and enumeration of cells were performed using Columbus image analysis software. Only compounds that were not toxic (cell viability >75%, in blue in the upper panel) were kept for further analysis. The number of live cells and the areas of intracellular bacteria ($px^2$: pixels$^2$, lower panel) were expressed as the percentage in compound-treated cells compared to cells incubated with DMSO. **(B)** Representative confocal images of macrophages infected with MTB (green) and treated

with MC3465. Hoechst 33342 and HCS CellMask (blue) were used to delimit nuclei and cytoplasm shapes, respectively. Scale bar: 10 μm. **(C)** MTB-infected macrophages were treated with MC3465 (1 and 10 μM). The number of intracellular bacteria was enumerated at 0, 24, and 96 h postinfection. One representative experiment (of at least 3) is shown. **(D)** Macrophages were infected with clinical strains of MTB, namely, GC1237, CDC1551, and Myc5750 and were treated with MC3465 (10 μM). The number of intracellular bacteria was enumerated at 96 h posttreatment. **(E)** Macrophages were infected with *L. monocytogenes* or *S.* Typhimurium. Intracellular bacteria were enumerated at 0 and 24 h postinfection. **(F)** Growth of MTB in culture liquid medium in the presence of MC3465 (10 and 50 μM). Data are representative of 2 independent experiments. **(G)** MTB-infected macrophages, MTB separated from macrophages using Transwell inserts, and MTB alone were treated with MC3465 (10 μM) or DMSO for 96 h. Cells were lysed and the number of intracellular bacteria was enumerated. Panels D, F, and G: Data are representative of 2 independent experiments. Error bars represent the mean ± SD. One-way ANOVA test (1C) and *t* test (1D, 1G) were used. ns, not significant, * $p < 0.05$, ** $p < 0.01$, *** $p < 0.001$, **** $p < 0.0001$. The data underlying the graphs shown in the figure can be found in S1 Data.

were cultivated (i) within macrophages; (ii) in presence of macrophages, without any direct contact with the host cells; or (iii) alone. After 96-h treatment, the number of bacteria was enumerated in each condition. As shown in Fig 1G, MC3465 only limits the multiplication of intracellular MTB. This observation suggests that MC3465 is not metabolized by macrophages in an active form that could target extracellular bacteria.

## MC3465 inhibits MTB growth in a SIRT2-independent manner

MC3465 or 3-(4-bromophenyl)-5-(3-bromopropyl)-1,2,4-oxadiazole is described to be a specific human SIRT2 inhibitor, with no activity against SIRT1, SIRT3, and SIRT5 [20]. It has been published that the expression of SIRT2 is up-regulated upon MTB infection and that inhibition of SIRT2 by AGK2 leads to a decrease of intracellular MTB survival in Raw 264.7 macrophages and murine peritoneal macrophages [21]. To assess if MC3465 limits MTB intracellular growth through SIRT2 inhibition, we incubated MTB-infected macrophages with either DMSO, MC3465, or with the 2 commonly used human SIRT2 inhibitors: AGK2 and SirReal2 [22]. Cells were lysed 96 h postinfection, and the number of bacteria was determined by CFUs. While MC3465 decreased the number of viable intracellular MTB in a dose-dependent manner, AGK2 and SirReal2 had no effect on bacterial growth, suggesting MC3465 acts in a different way to the SIRT2 inhibitors (S3A and S3B Fig). It may also suggest that SIRT2 plays a different role in human and mouse macrophages during MTB infection. To keep investigating if SIRT2 is the putative target of MC3465, we inactivated SIRT2 expression in human macrophages using siRNA-mediated gene silencing [23]. The level of SIRT2 expression decreased by about 70% upon silencing (S3C Fig). In SIRT2-silenced cells, MC3465 restrained the intracellular multiplication of MTB as well as in cells expressing normal level of SIRT2 (S3D Fig). Moreover, MC3465 was still active on intracellular MTB in mouse SIRT2$^{-/-}$ embryonic fibroblasts, to a similar level as the wild-type (WT) cells (S3E Fig). Altogether, our results strongly demonstrate that SIRT2 is not the target of MC3465. They also argue against a role of SIRT2 in the interactions between MTB and human macrophages. The role of SIRT2 in MTB infection remains thus unclear, with additional complexities highlighted by Cardoso and colleagues [24], who found that SIRT2 deletion in the myeloid lineage transiently increased MTB load in the lungs and liver of conditional mice but did not impact long-term infection.

Sirtuins are a family of NAD$^+$-dependent lysine deacetylases highly conserved from bacteria to humans. MTB has a NAD$^+$-dependent protein deacetylase, namely, Rv1151c [25]. While Rv*1151c* is a nonessential gene for in vitro growth of MTB [26], we cannot exclude a role in macrophage parasitism. We generated a Rv*1151c* deletion mutant in H37Rv (ΔRv*1151c*) (S4A Fig). As expected, there was no difference in growth between ΔRv*1151c* and the WT MTB strain in liquid culture medium (S4B Fig). We then infected macrophages with both strains and treated the cells with MC3465. After 96 h of infection, the cells were lysed and the number

of intracellular bacteria was enumerated. ΔRv*1151c* was able to grow intracellularly at a similar level as the WT strain and was as susceptible to MC3465 as the WT (S4C Fig). This result excludes an inhibition of the mycobacterial SIRT2-like protein by MC3465.

## MC3465 and its analog MC3466 alter zinc homeostasis, resulting in bacterial poisoning

Autophagy is a well-established key factor for host defense against MTB [27], and several autophagy activating drugs have been used to restrict MTB survival [4]. We therefore tested whether MC3465 activated the formation of autophagosomes and autolysosomes. MTB-infected macrophages were stained with an antibody against microtubule-associated protein light-chain 3B (LC3B), and the fluorescence was analyzed by confocal microscopy. We observed minimal difference in the number of LC3B puncta per cell at 4 h and 48 h posttreatment (S5A and S5B Fig). On the contrary, we observed a decrease in the number of puncta per cell 4 h after treatment with MC3465. This decrease was no longer observed after 48 h of treatment. During autophagy, LC3 undergoes a series of modifications, leading to the generation of 2 forms: LC3-I and LC3-II. Once lipidated, the cytoplasmic form LC3-I is converted to LC3-II and associates with the autophagosomal membrane. We therefore investigated the presence of these 2 forms of LC3 in MC3465-treated cells. Western blot analysis revealed no difference between DMSO- and MC3465-treated cells (S5C Fig). Bafilomycin A1 (BAF) was used as a control. It is a specific inhibitor of the vacuolar-type H+-ATPase (V-ATPase), which blocks the acidification of lysosomes and prevents the fusion of autophagosomes with lysosomes. As a result, LC3-II accumulates within the autophagosomes, leading to an increase in its intracellular levels. As expected, we observed an increase of LC3-II in BAF-treated cells during 4 and 48 h. To confirm this data, we used well-characterized autophagy inhibitors such as 3-methyladenine (3-MA), BAF, or chloroquine (CQ) and tested their effect on MTB-infected macrophages treated with MC3465. Intracellular bacteria were collected and plated to enumerate the CFUs 96 h postinfection. Our data show that autophagy inhibitors 3-MA, BAF, or CQ do not inhibit the effect of MC3465 (S5D Fig). MC3465 activity is thus not related to an induction of autophagy.

Reactive oxygen species (ROS) are fundamental for macrophages to eliminate pathogens. We thus evaluated investigated whether MC3465 induces ROS production in both mitochondria and the broader cellular environment. ROS were visualized using 2 different fluorescent probes: MitoSOX and CellROX. MC3465 did not induce the production of ROS in MTB-infected macrophages at 4 and 24 h posttreatment (S6A–S6D Fig). As expected, treatment with the antioxidants, glutathione (GSH) and its precursor N-Acetyl-L-cysteine (NAC), did not inhibit the efficacy of MC3465 (S6E Fig).

To further understand the molecular mechanism of action of MC3465 to restrain intracellular MTB growth, we performed a transcriptome analysis upon MC3465 treatment. We infected macrophages from 4 healthy donors with MTB and then treated them with MC3465. After 4 h and 24 h treatment, we characterized the genome-wide gene expression profiles of macrophages by RNA-seq, with DMSO-treated cells serving as a control. The expression of 181 genes was differentially regulated following MC3465 treatment ($p$-value $< 0.05$, log FC $\geq 0.5$ and $\leq -0.5$) after 4 h, with 53 being up-regulated and 128 being down-regulated (Fig 2A). The expression of more genes was affected after 24 h treatment. A total of 224 genes were up-regulated and 652 genes down-regulated ($p$-value $< 0.05$, log FC $\geq 0.5$ and $\leq -0.5$) (Fig 2A and S2–S5 Tables).

We classified the differentially expressed genes by performing gene-set enrichment analysis using ClueGO cluster analysis [28]. The gene set down-regulated by MC3465 at 4 h was

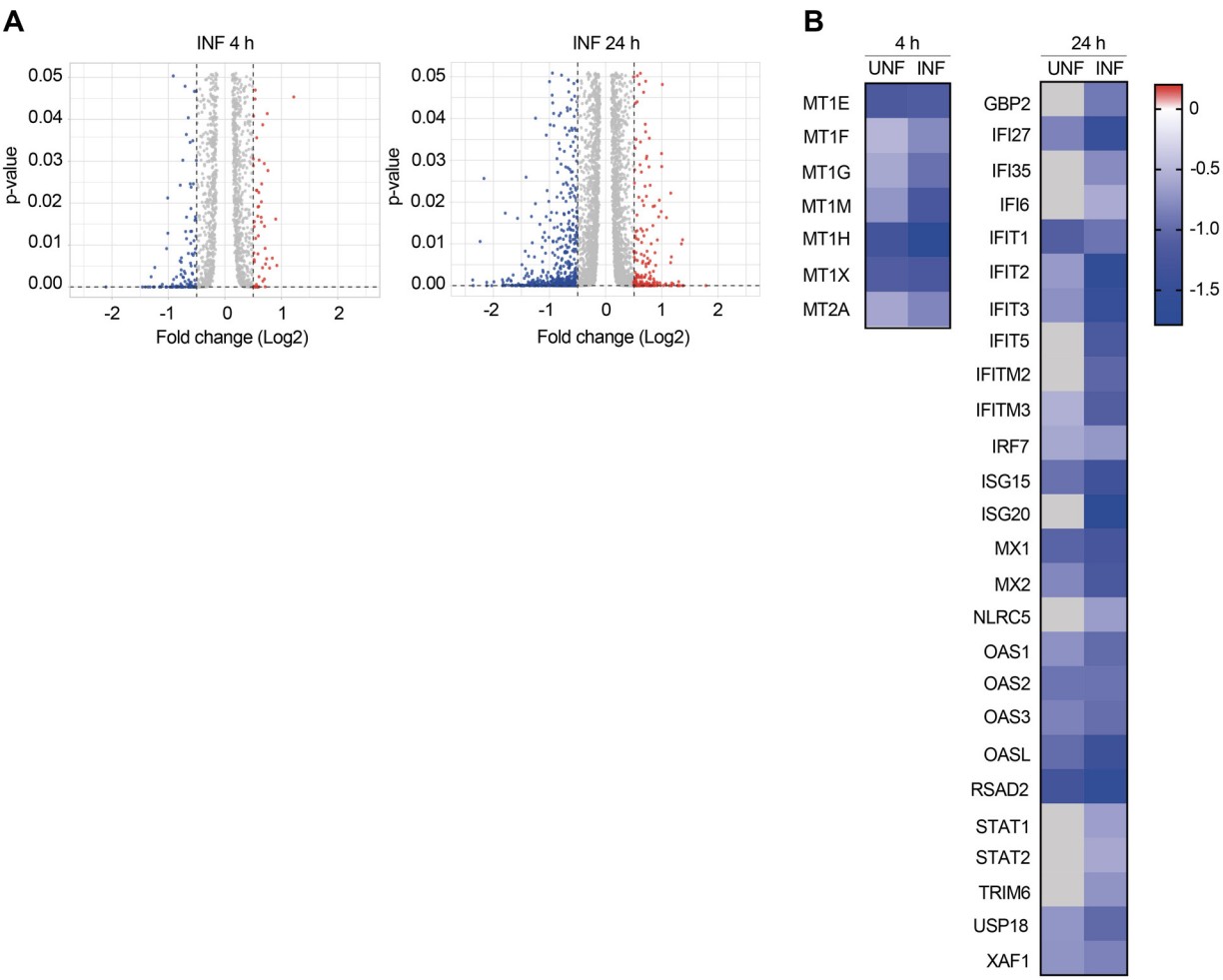

**Fig 2. Differentially expressed genes upon MC3465 treatment.** Naive and MTB-infected macrophages derived from 4 individual donors were treated with MC3465 (10 μM) for 4 h and 24 h. Differentially expressed genes were identified by mRNAseq. **(A)** Volcano plot showing differentially expressed genes due to MC3465 treatment in MTB-infected cells (*p*-value <0.05, Log Fold Change (FC) <−0.5 and >0.5). **(B)** Heatmap showing expression of genes differentially expressed in naive (UNF) and MTB-infected cells (INF) upon MC3465 treatment. Genes related to cellular zinc ion homeostasis and type I IFN signaling pathway were represented at 4 and 24 h, respectively. Genes that were not differentially expressed were represented by a grey square.

significantly enriched for genes associated with chemotaxis and cellular zinc ion homeostasis and cell chemotaxis (Table 1 and Figs 2B and S7A). The limited number of genes up-regulated by MC3465 hindered the identification of specific GO categories. At 24 h, most of the down-regulated genes belongs to the innate immune response and the cytokine response, with many genes belonging to the type I interferon (IFN) pathway (Figs 2B and S7B).

Maintaining intracellular zinc homeostasis is essential for all domains of life, as zinc serves as cofactor for enzymes and other proteins involved in different fundamental cellular processes. In this context, nutritional immunity is a well-known mechanism deployed by the host cell to limit the availability of essential metals to starve pathogens and, hence, impedes the intracellular growth of the pathogens [29]. In contrast, intracellular bacteria can be intoxicated by an excess of free metal ions, namely, zinc and copper [30,31]. Our transcriptomic data in MTB-infected and naive macrophages suggested that MC3465 alters zinc sequestration. The expression of genes coding for metallothioneins was down-regulated upon treatment (Fig 2B).

**Table 1. Gene ontology (GO) enrichment analysis of genes whose expression is down-regulated by MC3465 treatment in MTB-infected cells.**

| 4 h p.i. | | | |
|---|---|---|---|
| GO category | Term | *p*-value | Bonferroni adjusted *p*-value |
| GO:0050922 | Negative regulation of chemotaxis | $2.11 \times 10^{-7}$ | $7.60 \times 10^{-6}$ |
| GO:2000273 | Positive regulation of signaling receptor activity | $2.37 \times 10^{-4}$ | $4.97 \times 10^{-3}$ |
| GO:0006882 | Cellular zinc ion homeostasis | $1.03 \times 10^{-10}$ | $4.22 \times 10^{-9}$ |
| GO:0032640 | Tumor necrosis factor production | $9.86 \times 10^{-5}$ | $2.56 \times 10^{-3}$ |
| GO:0060326 | Cell chemotaxis | $2.56 \times 10^{-8}$ | $9.48 \times 10^{-7}$ |
| 24 h p.i. | | | |
| GO category | Term | *p*-value | Bonferroni adjusted *p*-value |
| GO:0000278 | Mitotic cell cycle | $7.45 \times 10^{-24}$ | $8.69 \times 10^{-21}$ |
| GO:1903047 | Mitotic cell cycle process | $7.26 \times 10^{-24}$ | $8.47 \times 10^{-21}$ |
| GO:0034097 | Response to cytokine | $4.92 \times 10^{-18}$ | $5.70 \times 10^{-15}$ |
| GO:0045930 | Negative regulation of mitotic cell cycle | $1.46 \times 10^{-15}$ | $1.67 \times 10^{-12}$ |
| GO:0006260 | DNA replication | $1.85 \times 10^{-19}$ | $2.15 \times 10^{-16}$ |
| GO:0051726 | Regulation of cell cycle | $1.80 \times 10^{-17}$ | $2.08 \times 10^{-14}$ |
| GO:0045087 | Innate immune response | $1.09 \times 10^{-21}$ | $1.27 \times 10^{-18}$ |
| GO:0035556 | Intracellular signal transduction | $3.45 \times 10^{-17}$ | $3.98 \times 10^{-14}$ |

Metallothioneins are cysteine-rich metal-binding proteins that are important for zinc and copper homeostasis, protection against oxidative stress, DNA damage, and heavy metals toxicity [32]. Their down-regulation in treated cells could, therefore, be associated with an increase of intracellular free zinc.

To test this hypothesis, naive and MTB-infected macrophages were treated with MC3465 during 4 h and stained with FluoZin 3-AM, a fluorescent $Zn^{2+}$-selective indicator [33,34]. Cells treated with the zinc-chelating cell-permeant agent TPEN [30] were used as control. TPEN treatment significantly decreased FluoZin 3-AM fluorescence to near background levels. Conversely, treatment with MC3465 resulted in a rapid increase in FluoZin 3-AM signal whether the cells were infected or not (Figs 3A and 3B, S8A and S8B). The labeling was as strong as in cells cultured in medium supplemented with high concentrations of $ZnSO_4$ (S8B Fig). This increase in free zinc staining persisted for up to 4 d (S8C and S8D Fig). Moreover, we also found numerous associations between the FluoZin 3-AM signal and DsRed-expressing MTB, indicating a zinc accumulation in bacteria-containing phagosomes after 4 h of MC3465 treatment (Fig 3C and 3D). We next tested whether MC3465 limited intracellular MTB growth through the influx of zinc into the phagosome. MTB-infected macrophages were treated with TPEN, which inhibited the increase in the FluoZin 3-AM signal ([30]; Fig 3A and 3B). After 1 h of treatment, cells were incubated with MC3465 and the number of bacteria was determined by CFUs at 96 h posttreatment. As shown in Fig 3E, TPEN abolished the effect of MC3465, demonstrating that zinc release is required for better control of MTB upon MC3465 treatment. The cells then became highly susceptible to MTB infection, regardless of whether the cells were treated with MC3465. This result suggests that zinc may be present in the phagosome at levels undetectable by FluoZin 3-AM in the absence of TPEN and that this may, in part, limit MTB growth.

MC3465 treatment may also modulate the intracellular levels of other metals such as Cu. MTB-infected cells were treated with MC3465. After 4 and 24 h of treatment, cells were labeled with the BioTracker Green copper dye. This is a fluorescent probe that detects Cu+, which is a

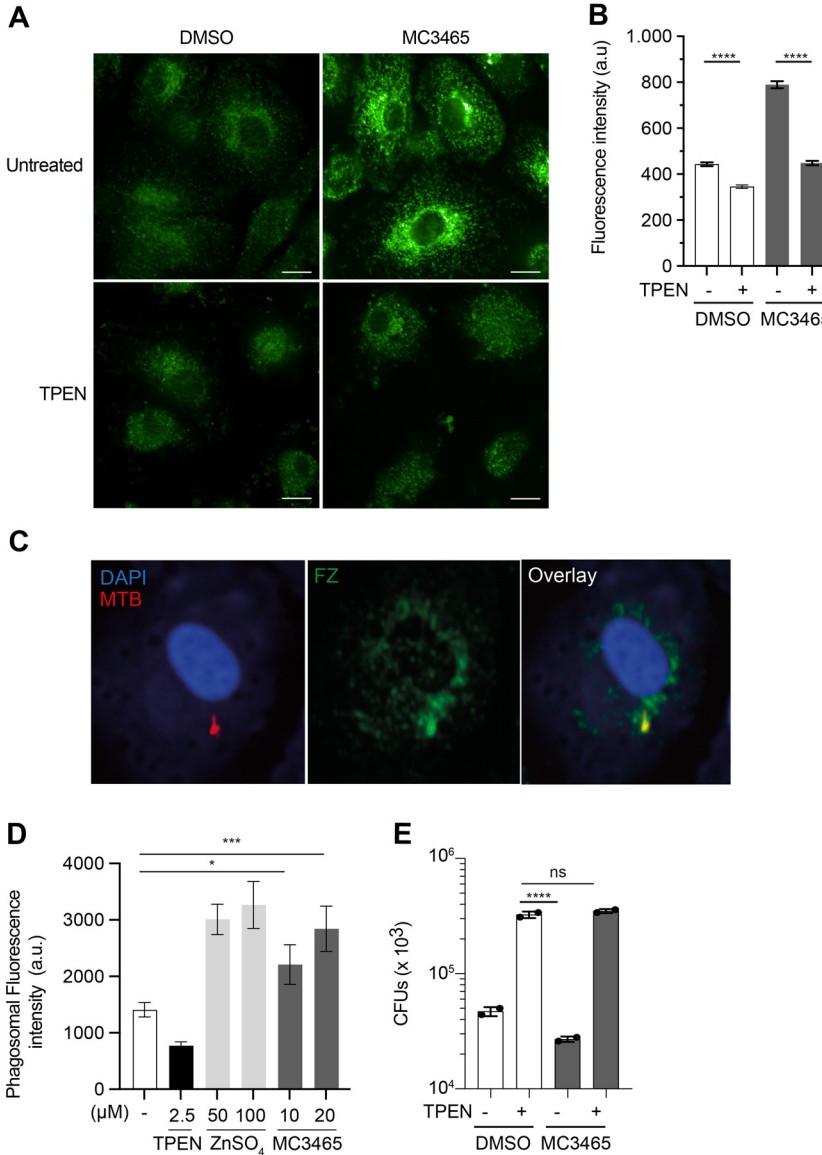

**Fig 3. Free zinc is released upon MC3465 treatment and accumulates within the mycobacterial phagosome. (A)**
Macrophages were infected with MTB during 2 h and incubated with the zinc-chelating agent TPEN (2.5 μM). After 1 h, cells were treated with MC3465 for 4 h. Cells were fixed and stained with the free zinc-specific fluorescent probe FluoZin 3-AM. Scale bar, 10 μm. **(B)** Average FluoZin 3-AM signal intensity, for at least 400 cells per condition. **(C, D)** Macrophages were infected with DsRed-MTB and incubated with TPEN, ZnSO$_4$ (100 μM), or MC3465 (10 μM) during 4 h. Cells were then fixed and stained with FluoZin 3-AM. Data are representative of 2 independent experiments. **(C)** Representative image of FluoZin 3-AM association with MTB. **(D)** Quantification of FluoZin 3-AM association with DsRed-MTB. At least 100 phagosomes were counted per condition. **(E)** MTB-infected cells were incubated with TPEN (2.5 μM). After 1 h, cells were treated with MC3465 (10 μM). The number of intracellular bacteria was enumerated at 96 h posttreatment. Data are representative of 3 independent experiments. One-way ANOVA test was used. Error bars represent the mean ± SD. ns, not significant, * $p < 0.05$, *** $p < 0.001$, **** $p < 0.0001$. The data underlying the graphs shown in the figure can be found in S2 Data.

dominant redox state of copper in an intracellular reducing environment. We observed no change in staining by MC3465 compared to DMSO (S9A and S9B Fig).

It has been shown that zinc exposure leads to rapid induction of the putative heavy metal efflux P1-type ATPase ctpC (Rv3270) [30]. We therefore analyzed the expression of *ctpC* by

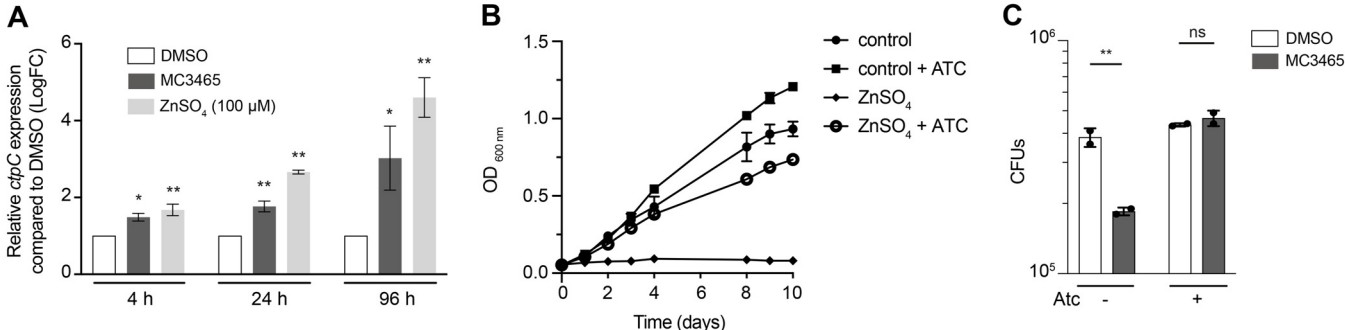

**Fig 4. The mycobacterial P$_{1B}$-ATPase metal exporter, CtpC, along with the chaperone-like protein PacL1, reverses zinc intoxication mediated by MC3465. (A)** RT-qPCR analysis of *ctpC* expression upon incubation with MC3465 (10 μM) or ZnSO$_4$ (100 μM). After 4, 24, and 96 h treatment, total cellular RNA was extracted and analyzed by RT-qPCR. Data are normalized relative to *ftsZ* gene. **(B)** Growth of the triple *ctp* mutant, expressing P$_{pacL1}$-driven CtpC and anhydrotetracycline (Atc)-inducible PacL1, was observed in liquid culture medium supplemented with ZnSO$_4$ (200 μM) and Atc (200 ng/ml). **(C)** MTB-infected macrophages were treated with either DMSO or MC3465 (10 μM) in the presence or absence of Atc (200 ng/ml). The number of intracellular bacteria was enumerated at 96 h posttreatment. Data are representative of 2 independent experiments. Error bars represent the mean ± SD. One-way ANOVA test was used. ns, not significant, *$p < 0.05$, **$p < 0.01$. The data underlying the graphs shown in the figure can be found in S3 Data.

quantitative reverse transcription PCR (RT-qPCR) in cells incubated with MC3465 or the positive control ZnSO$_4$ during 4, 24, and 96 h. The expression of *ctpC* was increased in response to both compounds (Fig 4A).

CtpC colocalizes with PacL1, and the 2 proteins form high molecular weight complexes [35]. Evidence indicates that the presence of PacL1 is essential for maintaining high levels of CtpC and resistance to zinc intoxication [35]. A MTB strain with elevated expression levels of both PacL1 and ctpC should thus exhibit decreased sensitivity to MC3465. To test this hypothesis, we generated a triple mutant in which the 3 operons *pacL1* (Rv3269)-*ctpC*, *pacL2* (Rv1993c)-*ctpG*, and *pacL3* (Rv0968)-*ctpV* have been deleted and which expressed P$_{pacL1}$-driven CtpC and anhydrotetracycline (Atc)-inducible PacL1. As expected, the triple mutant was sensitive to zinc at 200 μM (Fig 4B), and zinc resistance was restored in the presence of Atc (Fig 4B). We then infected macrophages with this strain and treated the cells with DMSO or MC3465 in the presence or absence of Atc. After 96 h of infection, the cells were lysed, and the number of intracellular bacteria was enumerated. In the absence of Atc, MC3465 inhibited the growth of MTB. However, in the presence of Atc, the mutant was able to grow intracellularly at a level similar to bacteria treated with DMSO (Fig 4C). Overall, these results indicated that MTB is exposed to a high concentration of zinc during treatment with MC3465.

## Structure–activity relationship study of MC3465 allowed the identification of MC3466, a more potent analog, effective against clinical strains of MTB

We then sought to develop a more effective yet endowed with low toxicity compound such as MC3465. Structure–activity relationship study is a key aspect of any drug design and development campaign [36]. MC3465 is a 1,2,4-oxadiazole derivative bearing a 4-bromophenyl moiety at C3 and a 3-bromopropyl chain at C5. We tested 29 analogs of MC3465 (structures in S6 Table) for their ability to restrain MTB growth within human macrophages, as described in Fig 1. These compounds possess different substitutions on the phenyl moiety at C3 and alkyl, haloalkyl, alkoxy, alkylaryl, or carboxamide chains at the C5 position, with 2 of them also having central cores different from the original 1,2,4-oxadiazole one. Using high-content screening confocal microscopy, we identified active and inactive analogs of MC3465 (Fig 5A and S6 Table). This approach allowed us to identify the presence of an electron-withdrawing group at

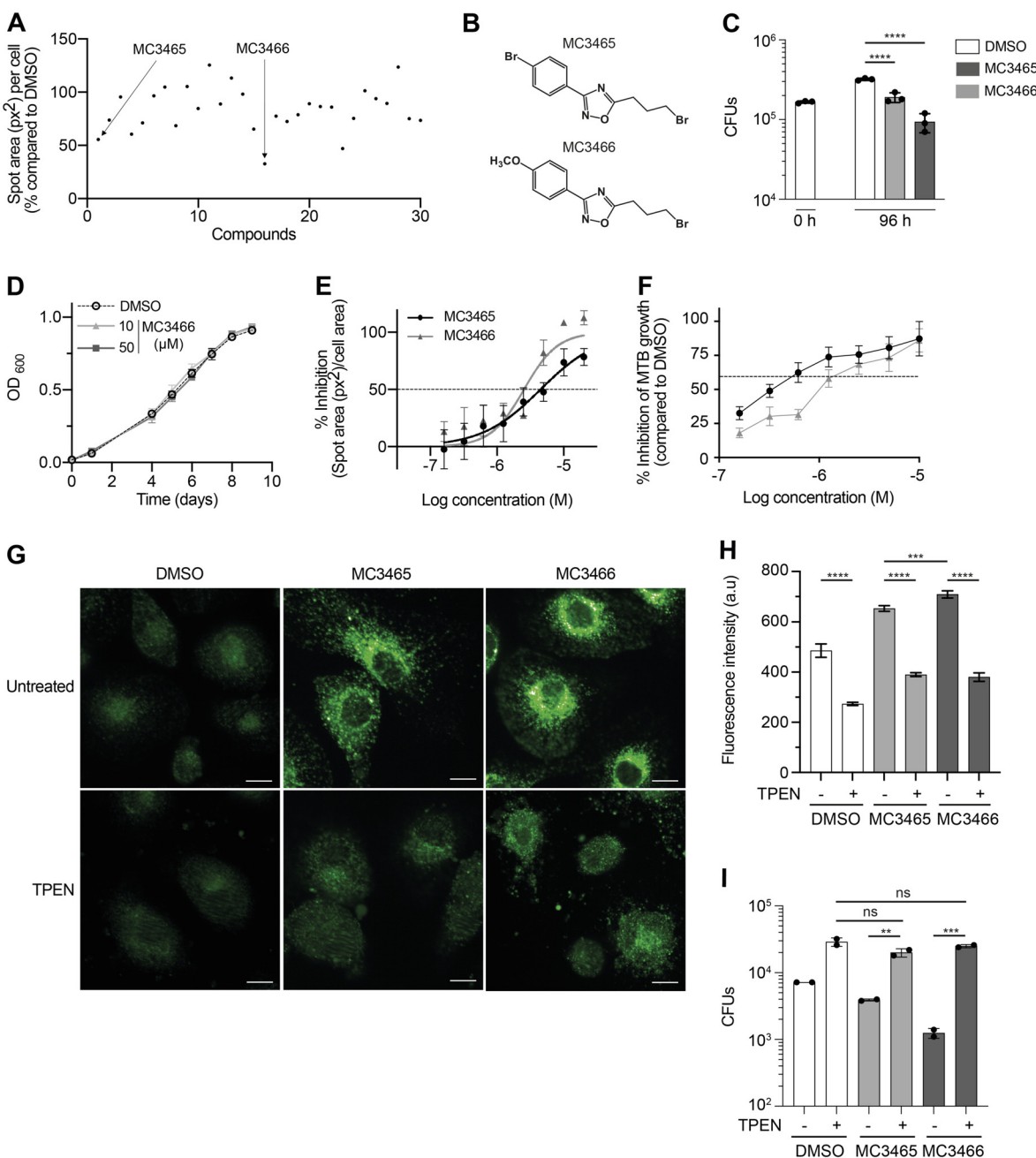

**Fig 5. Structure–activity relationship analysis of MC3465 and identification of MC3466 as a more effective analog. (A)** Macrophages were infected with GFP-MTB and incubated with 29 MC3465 analogs (10 μM). Images were acquired by automated confocal microscopy followed by image analysis 96 h postinfection, as described in Fig 1A and 1B. The spot area per cell was expressed as the percentage of GFP area in compound-treated cells compared to cells incubated with DMSO. **(B)** Structures of MC3465 and MC3466. **(C)** MTB-infected macrophages were treated with MC3465 or MC3466 (10 μM). After 96 h, the number of intracellular bacteria was enumerated. One representative experiment (of at least 3) is shown. **(D)** MTB growth in liquid culture medium in the presence of the analog MC3466 at different concentrations, determined by $OD_{600}$. Data are representative of 3 independent experiments. **(E)** The dose–response curves (DRCs) for MC3465 and MC3466 were obtained by automated confocal microscopy followed by image analysis. The ratio of GFP area in compound-treated macrophages, compared to cells incubated with DMSO, was normalized with the negative control DMSO (0% inhibition) and the positive control RIF (100% inhibition). **(F)** MTB-infected cells were treated with different concentrations of MC3465 or MC3465 (10 μM). After 96 h, bacteria were enumerated by CFU and the IC50 of each compound was determined. **(G-I)** MTB-infected macrophages were incubated with TPEN (2.5 μM). After 1 h, cells were treated with MC3465 or MC3466 (10 μM). **(G)** Representative image of FluoZin 3-AM staining at 4 h posttreatment. Scale bar, 20 μm. **(H)** Average FluoZin 3-AM signal intensity for at least 400 cells per condition. **(I)** The number of intracellular bacteria was enumerated at 96 h posttreatment. Panels E and F: Data are representative of 2

independent experiments. One-way ANOVA test was used. Error bars represent the mean ± SD. ns, not significant, ** $p < 0.01$, *** $p < 0.001$, **** $p < 0.0001$. The data underlying the graphs shown in the figure can be found in S4 Data.

the para position of the C3-phenyl ring as very important for compound activity. Furthermore, the 3-bromopropyl chain guaranteed the best compound activity, and switching to other kinds of side-chains was detrimental, except for compound MC3617, which has a carboxamide on C5.

Interestingly, we identified MC3466 as the most effective molecule of this series, with higher potency than MC3465, and the capability to reduce the area of GFP-expressing bacteria by 69% compared to the control (Fig 5A). Although the structures of MC3466 and MC3465 are very similar, MC3466 differs from MC3465 for the presence of a methoxy group, rather than a bromine atom, at the *para* position of the C3-phenyl ring (Fig 5B). We confirmed this result by counting the number of bacteria inside macrophages treated with MC3466 for 96 h (Fig 5C). MC3465 and MC3466 decreased MTB growth by 50% and 67%, respectively. As MC3465, the analog MC3466 did not affect cell viability and had no effect on the growth of MTB in liquid culture medium at 10 μM nor at a higher concentration (50 μM) (Figs 5D and S10A). We next assessed the half-maximal inhibitory concentration (IC$_{50}$) of both MC3465 and MC3466. Briefly, macrophages were infected with MTB and treated with a range of concentration of MC3465 and MC3466 for 96 h. The IC$_{50}$ was estimated using CFU-based assay or image-based analysis with dose–response curve (DRC) as previously described [37]. We obtained similar results with these 2 methods, with estimated IC$_{50}$ values for MC3465 of 4.5 or 5.9 μM and 2.5 or 3.2 μM for MC3466, respectively (Fig 5E and 5F). As described for MC3465, MC3466 induced an increase in free zinc in macrophages (Fig 5G and 5H). Inhibition of this release by TPEN cancels out the compound's effects on intracellular MTB growth (Fig 5I). As a control, no zinc release was induced in cells treated with MC3465 analog, which do not inhibit intracellular growth of MTB (S10B–S10E Fig).

## MC3466 potentiates the activity of known anti-TB drugs

HDT aims to increase the success of TB treatment. However, HDT agents are more likely to be used in addition to conventional anti-TB drugs that directly target the bacteria [4]. Our data have shown that MC3465 and MC3466 help human macrophages to control the growth of MTB but are insufficient to eradicate the bacteria. We therefore investigated whether MC3466 could act together with some anti-TB drugs to improve the efficacy of the treatment. We first tested the association between MC3466 and the anti-TB drugs, RIF, and bedaquiline (BDQ) in liquid culture medium. As expected, we showed no synergy between MC3466 and RIF or BDQ on the growth of MTB (S11A–S11C Fig). We then focused on MTB-infected macrophages treated with MC3466 and different concentrations of the 2 drugs. After 96 h treatment, cells were lysed and bacteria counted. As expected, both anti-TB drugs alone induced a decrease in bacterial numbers compared to untreated cells in a dose-dependent manner (Fig 6A and 6B). However, in the presence of MC3466, RIF and BDQ showed a higher bactericidal activity. At low concentration of the 2 antibiotics (0.1 μg/ml), MC3466 increased the efficiency of RIF by 93% and BDQ by 97%. This effect seems to go far beyond an additional effect of the compound and these anti-TB drugs; MC3466, RIF, and BDQ alone reduced the bacterial load by 65%, 65%, and 79% compared to DMSO, respectively (Fig 6A and 6B).

We next used the Bliss independence model to confirm these results. This model is widely used to determine independence, synergy, or antagonism between 2 compounds [38]. It provides a theoretical framework to predict the expected effect of combining 2 drugs based on

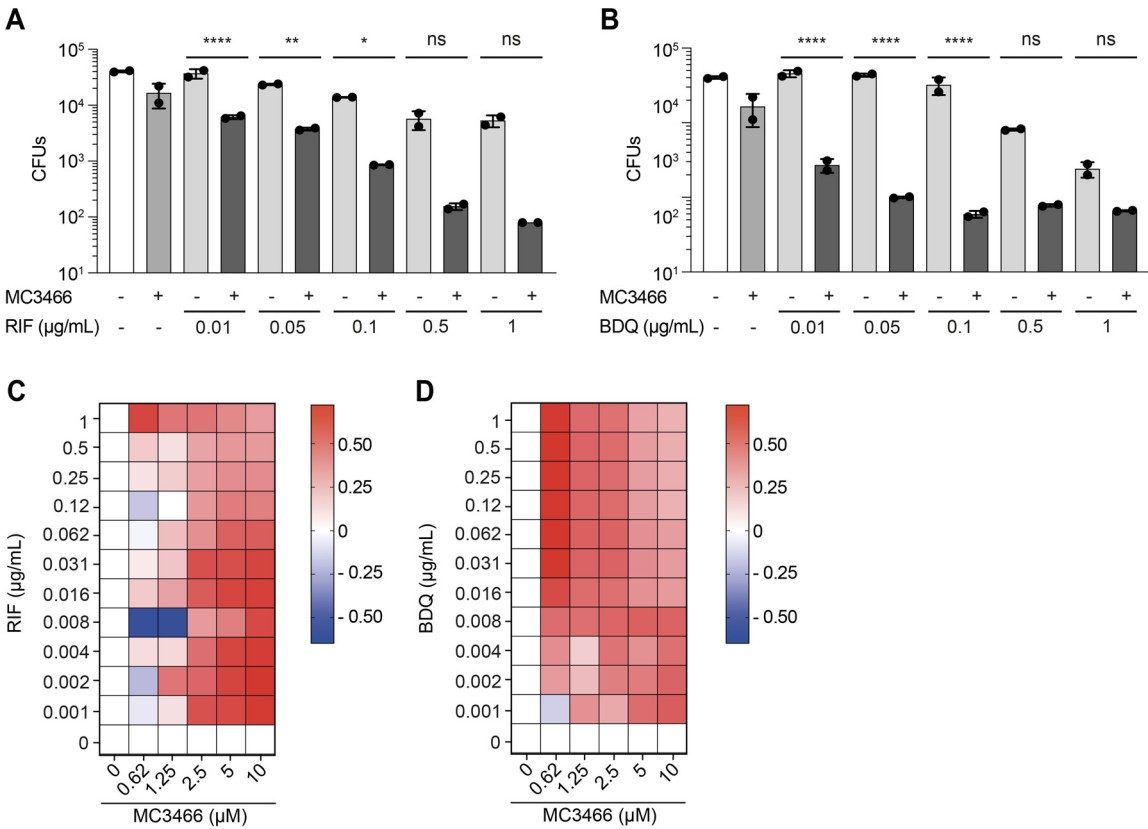

**Fig 6. MC3466 potentialized the efficacy of rifampicin and bedaquiline in MTB-infected macrophages. (A)** Macrophages were infected with the drug-susceptible MTB H37Rv and treated with various concentrations of RIF in association or not with MC3466. After 96 h, cells were lysed and the number of intracellular bacteria enumerated. **(B)** As in (A) except that RIF has been replaced by BDQ. **(C, D)** Heatmaps representing variation in drug combination, using the Bliss independence model, ranging from antagonism (blue) to synergy (red). Cells were infected with GFP-MTB and were treated with a range of concentrations of MC3466 and RIF or BDQ. Images were acquired by automated confocal microscopy followed by image analysis 96 h posttreatment, as described in Fig 1. Panels A and B: One representative experiment (of at least 3) is shown. Panels C and D: One representative experiment (of 2) is shown. One-way ANOVA test was used. Error bars represent the mean ± SD. ns, not significant, *$p < 0.05$, **$p < 0.01$, ****$p < 0.0001$. The data underlying the graphs shown in the figure can be found in S5 Data.

their individual effects when used alone. The model assumes that the drugs act independently, and any deviation from the predicted effect may indicate synergy (greater-than-expected effect) or antagonism (less-than-expected effect). A positive Excess over Bliss score indicates synergy. Conversely, a negative Excess over Bliss score indicates antagonism. MTB-infected macrophages were treated with a wide concentration range of MC3466 alone or associated with different concentrations of RIF or BDQ. After 96 h treatment, intracellular GFP-expressing MTB growth was assessed by determining the GFP area per cell. We confirmed that the combination of MC3466 with RIF or BDQ was highly bactericidal on MTB. This effect was not the result of an additive effect between the 2 drugs, but of a synergy (Fig 6C and 6D).

## MC3466 potentiates the efficiency of RIF in the mouse model of TB

Many drugs with high in vitro efficacy fail to produce significant effects in vivo. Many factors affect their absorption, distribution, or metabolism. Before testing our compounds in vivo in mice, we checked whether our molecules allowed murine macrophages to control MTB

infection. Raw 264.7 cells (a murine macrophage cell line) were infected with MTB and treated with both MC3465 and MC3466. After 48 h, cells were lysed and bacteria were enumerated by CFU. Both molecules were equally active in murine and human cells (S12A Fig).

We next tested the toxicity of MC3465 and MC3466 in naive mice. We injected intraperitoneally MC3465 and MC3466, 6 d a week for 2 wk in uninfected C57BL/6J mice. No loss of weight and abnormal behavior was observed during treatment (S12B Fig). After 2 wk, total blood was collected to quantify the level of liver enzymes, which might indicate inflammation or hepatocyte injury. No difference between the groups treated with the compounds and the control group was observed for the level of alanine aminotransferase (ALT) and aspartate aminotransferase (AST) (S12C and S12D Fig). These results suggest that both MC3465 and MC3466 treatments are well tolerated in vivo.

Our in vitro results showed that our compound enabled macrophages to fight MTB. We therefore wanted to assess efficacy in a mouse model where innate immunity plays a key role in controlling infection. We tested our molecule in MTB-infected mice in an acute infection model, alone or in combination with RIF. C57BL/6J mice were infected via the aerosol route. After 1 wk, mice were treated for 2 wk, 6 d a week with MC3466 with or without RIF. Lungs were harvested, and the bacterial load was assessed by CFUs (Fig 7A). On average, the number of bacteria after treatment with MC3466 was reduced by 5 times, although we observed important heterogeneity among treated mice. As observed in vitro, the combination of RIF and MC3466 was the most effective in decreasing the bacterial load (Fig 7A). The bacterial load was 31 times lower in MC3466/RIF-treated mice than in RIF-treated mice.

Histopathological observations of MTB-infected mice treated with RIF, MC3466, or both molecules confirmed these results. We observed large differences between the different groups, especially regarding the severity and organization of inflammatory lesions (Fig 7B and 7C). Lung sections of MC3466- or RIF-treated mice displayed significant interstitial syndrome and alteration of the bronchiolar epithelium; lungs of RIF-treated mice were less inflammatory. However, lungs of mice treated with both MC3466 and RIF resembled the lungs of healthy mice, with only residual inflammation. These results are consistent with the levels of Il-1β found in lung lysates from treated mice. As shown in Fig 7D, treatment with MC3466 resulted in a reduction in IL-1β levels, although to a lesser extent than observed with RIF alone. The combination of RIF and MC3466 did not further reduce IL-1β levels compared to RIF alone. This reduction is likely attributed to a decrease in bacterial load rather than an anti-inflammatory effect of MC3466.

## Discussion

Widespread antimicrobial resistance poses a threat to public health that requires prompt and adaptable approaches. High-throughput screening has been extensively used to identify potential targets for antimycobacterial agents [39] or drug candidates that inhibit MTB growth [40,41]. However, this approach rapidly showed some limitations as drugs inhibiting MTB growth in culture medium may be ineffective on intracellular MTB. For instance, host drug-metabolizing enzymes and transporters may influence antimicrobial pharmacokinetics and pharmacodynamics, thereby impacting their efficacy and/or toxicity [42]. The development of resistance to these molecules is also only a matter of time. In contrast, HDT is instead less prone to bacterial resistance and could also be used to reduce disease severity and mortality. Increasing the natural resistance of macrophages, the primary cell target of MTB, is indeed a promising avenue in the context of TB. Here, using high-content imaging on human MTB-infected macrophages, we identified 1,2,4-oxadiazole-based compounds, with no effect on MTB growth in liquid culture, but able to inhibit the intracellular mycobacterial growth inside

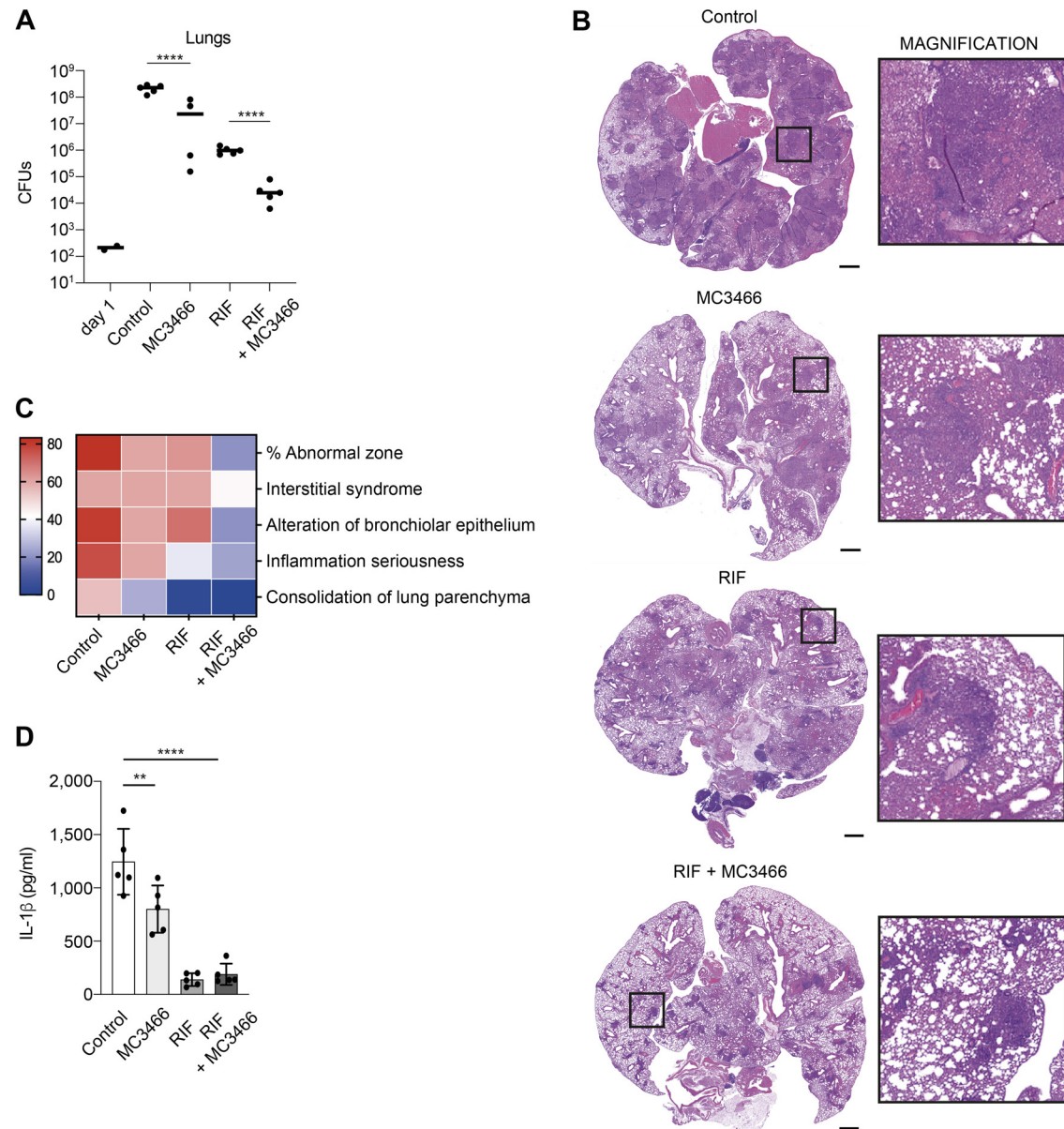

**Fig 7. Synergy between RIF and MC3466 in MTB-infected mice. (A)** C57BL/6J mice were infected by aerosol with MTB. After 1 wk, mice were treated for 2 wk, 6 d a week with MC3466 with or without RIF. Lungs were harvested, and the number of bacteria was estimated by CFU. **(B)** Representative hematoxylin and eosin stains of lungs, 2 wk after treatment. Scale bar: 1 mm. **(C)** Heatmap representing the histological scores upon RIF and MC3466 treatment. Lung sections are graded according to the percentage of abnormal area, presence of interstitial syndrome, alteration of bronchiolar epithelium, severity of inflammation, and consolidation of lung parenchyma. **(D)** Quantification of IL-1β in lung lysates from MTB-infected mice treated with MC3466 for 2 wk by ELISA. One representative experiment (of 2) is shown. Error bars represent the mean ± SD. One-way ANOVA test was used. **$p < 0.01$, ****$p < 0.0001$. The data underlying the graphs shown in the figure can be found in S6 Data.

the host in a concentration-dependent manner. Interestingly, these compounds have no similarity to other published 1,2,4-oxadiazole derivatives with activity on MTB [43–45].

Although the exact host target of MC3465 remains elusive, our data suggest that MC3465's mechanism of action is different from those previously described for HDTs. Indeed, most of the HDTs render macrophages less permissive to MTB [4,5] by favoring phagosome–lysosome

fusion, activating autophagy, or inducing antimicrobial peptides. Our data clearly showed that there is no activation of autophagy in presence of MC3465. However, we demonstrated that, upon treatment with MC3465, the expression of genes encoding metallothioneins was down-regulated in macrophages and associated with an increase of free zinc inside the cells and a relocation inside the phagosome, leading to MTB metal poisoning. This defense mechanism is commonly used by macrophages, overloading the phagosome with copper and zinc rendering the vacuolar environment unsuitable for bacterial survival [30,31]. Botella and colleagues showed that the expression of several metallothionein-encoding genes was induced upon MTB infection. It has been suggested that these genes were activated in response to the early burst of zinc inside the cells, aiming to decrease free zinc concentration and detoxify the cell. Our results indicate that treatment with MC3465 reduces metallothionein expression, preventing infected cells from efficiently eliminating the released free zinc. Metallothionein transcription is regulated by the zinc responsive transcription factor MTF-1 [46]. We did not observe any effect of MC3465 treatment on the expression and nuclear translocation of MTF-1. Metal-lothionein expression is also subject to regulation through epigenetic mechanisms, specifically DNA methylation of the promoter region, which can inhibit their expression [47,48]. It is tempting to speculate that MC3465 might interfere with the epigenetic mechanisms involved in metallothionein transcription.

MTB has developed different strategies to protect itself from metal poisoning [49]. For instance, following zinc intoxication from macrophages, it has been shown that MTB up-regu-lates the expression of genes encoding heavy metal efflux P-type ATPases CtpC, CtpG, and CtpV [30]. Our data show that MTB indeed increases the expression of *ctpC* in MC3465-treated macrophages. Additionally, we observed that an MTB strain with elevated expression levels of both PacL1 and CtpC displayed reduced sensitivity to MC3465. Altogether, our results demonstrate a novel mechanism of action of MC3465 as an HDT, related to zinc poisoning of the bacterial pathogen.

MC3465-treated macrophages control the growth of the laboratory MTB strain H37Rv and more virulent clinical MTB isolates, but they are still permissive to *L. monocytogenes* or *S.* Typhimurium infection. This result is likely due to the difference of lifestyle in the host cells between mycobacteria, *L. monocytogenes*, and *S.* Typhimurium. Upon uptake by macrophages, *L. monocytogenes* is engulfed in a phagosome but rapidly escapes to the cytosol where it multi-plies [50]. *S.* Typhimurium resides in a late endosome-like vacuole characterized by projection of filaments [51]. The maturation of the mycobacterial phagosome is arrested at an early stage of biogenesis by exclusion of the V-ATPase. Under certain circumstances, *S.* Typhimurium and MTB may also translocate into the cytosol [51]. Because of their very different intracellular lifestyle, MTB, *L. monocytogenes* and *S.* Typhimurium are thus likely exposed differently to dif-ferent zinc concentrations following MC3465 treatment. Moreover, MTB, *L. monocytogenes*, and *S.* Typhimurium have different sensitivity to zinc as they have developed their own strate-gies to sequester essential transition metals and protect themselves from toxicity. As an exam-ple, *Salmonella* appears to be less sensitive to the increase in cellular zinc levels following infection, as zinc supplementation and inactivation of metallothioneins 1 and 2 (which increased free zinc levels in macrophages), favor *Salmonella* survival [52]. Different intracellu-lar lifestyles associated to different resistances to zinc poisoning would explain the specificity of MC3465 towards mycobacterial strains. *E. coli* has been described as sensitive to intracellu-lar zinc. However, we did not observe any effect of MC3465 on *E. coli* survival in macrophages. It should be noted that macrophages are capable of efficiently eliminating *E. coli*, with almost no bacteria detected 24 h after infection. This rapid bacterial clearance may explain the lack of effect of our compound, as the macrophage response with the zinc burst was likely already optimized for efficient *E. coli* elimination.

In the fight against TB, there is growing interest in the development of effective antibacterial drug combinations for better therapeutic results [53,54]. Experimental drug regimens have been tested in vitro and optimized for their use in vivo [55]. Recently, drug combinations using an already approved drug with an adjuvant such as small molecules, have been exploited for the design of new drug regimens. These molecules are also called boosters, activators, or enhancers depending on the study [56]. For example, several 1,2,4-oxadiazoles were tested for their potency to boost the antibacterial activity of ethionamide [44,57]. HDT, associated to pathogen-targeted approaches, is a very promising approach for overcoming antimicrobial resistance. One of the striking findings of our study is the potentiation of classical anti-TB drugs, in vitro and in vivo, by the treatment of MC3465 or its analog MC3466. Indeed, we demonstrated that suboptimal concentrations of first- and second-line anti-TB drugs, RIF or BDQ, respectively, in combination with our compounds were more effective in preventing intracellular growth of MTB than those used alone. More importantly, this synergistic effect was also demonstrated in vivo for RIF in a mouse model of TB. In parallel, we also showed there is no weight loss nor indication of inflammation and hepatocyte injury, as the level of ALT and AST remains stable following 2 wk of treatment with our compounds. MC3466 was tested in an acute infection model. It would be interesting to test its efficacy in a chronic infection model, including various genetic backgrounds with different susceptibility phenotypes. The localization of the compound within the lesions could be studied by taking advantage of the bromine atom contained in the molecule, using an approach similar to that used for BDQ [58].

One of the possible applications of HDT in TB is to limit inflammation and tissue damage [5]. Histological analysis of lungs of MTB-infected mice treated with RIF, MC3466, or both revealed that association of RIF with MC3466 considerably reduced the number and severity of lesions. This is likely due to the greater efficacy of RIF when used in combination with MC3466 but may also be due to the down-regulation of the type I IFN pathway upon treatment with our compound. Indeed, our transcriptomic analysis of MC3465-treated macrophages showed indeed that genes involved in the type I IFN pathway are also down-regulated after 24-h treatment. Several studies have demonstrated that type I IFN overexpression is deleterious for the host during MTB infection [59]. In vitro, type I IFN and its downstream signaling cascade inhibited the antimicrobial response induced by type II IFNs in human monocyte [60]. In patients with active TB, blood transcriptional gene signatures with overexpression of type I IFN–related genes have been correlated with disease severity and are down-regulated following successful treatment [59,61]. Targeting the type I IFN pathway may be used as HDT. In the absence of IL-1, PGE2 failed to inhibit type I IFNs. The resulting overexpression of these cytokines leads to increased lung pathology. Administration of PGE2 and zileuton, a 5-LO inhibitor, negatively regulates type I IFN in vivo and confers protection in $Il1\alpha$, $Il1\beta^{-/-}$ mice infected with MTB [62].

In summary, we identified new compounds based on 1,2,4-oxadiazole scaffold, which allow human macrophages to control MTB replication and to potentiate the efficacy of 2 potent anti-TB drugs, namely, RIF and BDQ. These observations suggest that MC3465 and its analogs merit further investigation as potential HDTs.

## Materials and methods

### Ethics statement

Buffy coats were obtained from the French Blood Bank (Etablissement Français du Sang) under an agreement with the Institut Pasteur (C CPSL UNT, n˚12/EFS/134 and n˚15/EFS/023). Anonymous healthy donors signed a written informed consent to donate their blood for

research purposes. In accordance with French regulations, each blood donation is systematically screened for agents responsible for the following pathologies: AIDS, hepatitis B, hepatitis C, syphilis, and pathologies associated with HTLV. Latent TB infection is not tested. Animal studies were performed in agreement with European and French guidelines (Directive 86/609/CEE and Decree 87–848 of 19 October 1987). The study was approved by the Institut Pasteur Safety Committee (Protocol 11.245) and the ethical approval by local ethical committees "Comité d'Ethique en Experimentation Animale Institut Pasteur N° 89 (CETEA)" (CETEA dap 200037).

## Macrophages and cell lines

Blood mononuclear cells were isolated from buffy coats by Lymphocytes Separation Medium centrifugation (Eurobio). CD14+ monocytes were isolated by positive selection using CD14 microbeads (Miltenyi Biotec) and were cultured at 37°C and 5% $CO_2$ in RPMI-1640 medium (Gibco) supplemented with 10% heat-inactivated fetal bovine serum (FBS; Dutscher), and 2 mM L-glutamine (Gibco) with M-CSF (20 ng/mL; Miltenyi Biotec) (hereafter defined as complete medium). After 6 d of differentiation, the resulting macrophages were incubated in a buffer solution containing PBS, 2 mM ethylenediaminetetraacetic acid (EDTA), at 37°C and 5% $CO_2$ for 15 min before being harvested and counted.

THP-1 cell line was cultured in RPMI-1640 medium (Gibco) supplemented with 10% FBS and L-glutamine (Gibco). Mouse embryonic fibroblasts (MEFs) and Henrietta Lacks (HeLa, ATCC, CCL-2) cells were cultured in Dulbecco's Modified Essential Medium supplemented with L-glutamine (Gibco) and 10% FBS. Cells were incubated at 37°C, 5% $CO_2$, and the culture medium was changed every 2 d.

## Bacterial strains

*L. monocytogenes* strain WT EGD, number BUG600, was grown in brain–heart infusion medium (BHI; Difco Laboratories) at 37°C. *E. coli* (strain HB101) and *S. enterica* serovar Typhimurium (strain Keller) were grown in Luria Bertani (LB) medium. Mycobacteria were grown at 37°C in Middlebrook 7H9 medium (Becton-Dickinson) supplemented with 10% albumin dextrose catalase (ADC, Difco Laboratories) and 0.05% Tween 80. Hygromycin B was added to the culture medium of strains containing a plasmid encoding *gfp* or *dsred* gene. MTB strains used in this study include MTB strain H37Rv transformed with an Ms6-based integrative plasmid pNIP48 harboring GFP or DsRed protein [39], *M. bovis* BCG Pasteur, CDC1551, GC1237, Myc5750, and BDQ-resistant MTB strain H37Rv [63]. RIF-resistant MTB strain was provided by Anne-Laure Roux and Jean-Louis Herrmann (Assistance Publique Hôpitaux de Paris, Hôpital Ambroise Paré, France).

## Construction of MTB mutants

RvΔ*1151c* knock-out strain was obtained by performing DNA recombination. Briefly, 750-bp fragments of the upstream and downstream regions of Rv*1151c* gene were amplified by PCR from H37Rv genomic DNA. Primers used for Rv*1151c* upstream region are 5′-tgatgtaccta-caacccgaac-3′ and 5′-gactgagcctttcgttatttaaataatccgttcttgtcatcgcggaacgtcggt-3′. Primers used for Rv*1151c* downstream region are 5′-cgttccactgagcgatttaaattgatcgaagtcaatcccgagcccacgccg-3′ and 5′-ggtggattccacgaacgtgc-3′. The zeocin cassette was also amplified by PCR using primers 5′-atttaaataacgaaaggctcagtc-3′ and 5′-atttaaatcgctcagtggaacg-3′. A linear fragment composed of the zeocin cassette flanked by the Rv*1151c* upstream and downstream regions was amplified by PCR using 5′-aaaatatgatattcgcatggcg-3′ and 5′-aaagctcgaaagccgctggt-3′. This linear fragment was then electroporated into H37Rv harboring the pJV53-kanamycin plasmid,

previously grown in 7H9 medium supplemented with 0.2% acetamide for 24 h. After electroporation, 1 ml of 7H9 medium is added to the bacteria followed by an incubation step at 37°C for 48 h. Transformants are then plated on 7H11 containing both kanamycin and zeocin antibiotics. After 3 wk of incubation at 37°C, kanamycin-zeocin-resistant clones were screened by PCR using 3 different couples of primers: 5′-actacagctggatggattccg-3′ and 5′-atttaaatcgctcagtggaacg -3′ (amplification for if WT H37Rv: 0 bp, amplification if H37RvΔ$1151$c: 1531 bp), 5′-atttaaataacgaaaggctcagtc-3′ and 5′- acacgccagcgtcagcaatc-3' (amplification for if WT H37Rv: 0 bp, amplification if H37RvΔ$1151$c: 1857 bp), 5′-actacagctggatggattccg-3′ and 5′-acacgccagcgtcagcaatc-3′ (amplification for if WT H37Rv: 2,616 bp, amplification if H37RvΔ$1151$c: 2,550 bp). The Rv$1151$c knock-out gene was confirmed by sequencing.

Mutant strains of MTB H37Rv were constructed by allelic exchange using the recombineering technology [64], as described previously [65]. Strain H37Rv Δ*pacL1-ctpC*::*dif*6 Δ*pacL2-ctpG*::*dif*5 Δ*pacL3-ctpV*::*dif*4 is a triple mutant in which the 3 operons *pacL1* (Rv3269)-*ctpC*, *pacL2* (Rv1993c)-*ctpG*, and *pacL3* (Rv0968)-*ctpV* have been deleted from the chromosome and replaced by a 78-bp cassette harboring a variant of the chromosomal *dif* site [65]. This strain was transformed with plasmid pGMCS-TetR-P1-*pacL1*$_{flag}$-P$_{pacL1}$-*ctpC*$_{His6}$, an integrative vector expressing $P_{pacL1}$-driven his-tagged CtpC and anhydrotetracycline (Atc)-inducible flag-tagged PacL1 [35].

## Bacterial infection

Before infection, MTB were washed and resuspended in 4 mL PBS. Clumps were eliminated by passing through a 10-μm filter. The density of bacteria in the supernatant was verified by measuring the $OD_{600}$ and aliquot volumes defined to allow 1 bacterium per 2 cell infections. After 2 h of incubation at 37°C, infected cells were washed 3 times in PBS and treated with amikacin (50 μg/ml). After 1 h, cells were washed and incubated in fresh complete medium.

Infection of macrophages with the fast-growing bacteria *E. coli*, *L. monocytogenes*, and *S.* Typhimurium requires higher MOI values, consistent with the literature [66–70]. Briefly, *E. coli*, *L. monocytogenes*, and *S.* Typhimurium were grown to exponential phase. Bacteria were washed twice in PBS and added to cells at a MOI of 50:1. After 1 h, cells were washed in PBS, and gentamycin (50 μg/ml) was added to kill extracellular bacteria. After 1 h, cells were washed and incubated in fresh complete medium containing gentamycin (5 μg/ml).

## Compound synthesis

Compounds MC3465, MC3466, MC3209, MC3581, MC3582, MC3469, MC3453, MC3564, MC3565, MC3617, MC3459, MC3220, MC3212, MC4214, and MC4209 were synthesized as previously reported [15]. Full details regarding the synthesis and physico-chemical characterization of compounds MC3618, MC3586, MC3610, MC3577, MC3579, MC3735, MC3738, MC3775, MC3750, MC3748, MC3903, MC3904, MC3905, MC3573, and MC3578, will be reported elsewhere. All new compounds had spectral ($^1$H-NMR, ESI-MS) data in agreement with their chemical structures.

## Image-based high-content screening

Macrophages were infected with GFP- or DsRed-expressing H37Rv (MTB-GFP) and were cultured in 384-well plate (Cellcarrier plate, PerkinElmer); each well containing a compound resuspended in DMSO (0.1%) at a final concentration of 10 μM. Each compound was tested in triplicates. DMSO (0.1%) and RIF (1 μg/mL) were used as negative and positive controls, respectively. Cells were fixed with 4% paraformaldehyde at room temperature (RT) 96 h post-infection. After 1 h, cells were washed and stained with HCS CellMask Blue stain (2 μg/mL,

Thermo Fisher) and Hoechst 33342 (5 μg/mL, Thermo Fisher) in PBS for 30 min at RT, washed twice with PBS, and stored at 4°C until acquisition. Images were acquired using the automated confocal microscope Opera Phenix High-Content Screening System (Perkin Elmer Technology) with a 40×/NA 1.1 water objective followed by image analysis (Columbus Conductor Database, Perkin Elmer Technologies). The association between dsRed and FluoZin 3-AM signals was performed using the "Find Spots" building block. The intensity of the FluoZin 3-AM signal was then calculated within these spots. Data analysis was carried out to obtain numbers of cells, percentages of infected cells, and quantified areas of intracellular bacteria ($px^2$: $pixels^2$).

### Cell viability assay

Cell viability was determined using the MTT assay kit (Abcam), according to manufacturer's instructions.

### Resazurin assay determination of the minimal inhibitory concentration (MIC)

The microdilution test was performed in 96-well plates as previously described [63,71]. Briefly, MTB were cultured in 7H9 liquid medium containing 2-fold dilutions of antibiotics during 6 d. The dye resazurin (Sigma-Aldrich) at 0.02% (wt/vol) was then added to each well. After 24 h, the absorbance was measured at 570 nm.

### Determination of bacterial counts

Macrophages were lysed in distilled water with 0.1% Triton X-100. The number of bacteria was determined by plating 10-fold serial dilutions of the lysate, in triplicate, onto 7H11 agar plates. CFUs were scored after 3 wk at 37°C. *L. monocytogenes* and *S.* Typhimurium were plated on BHI and LB agars, respectively. CFUs were counted after 24 h at 37°C.

### Indirect immunofluorescence

Macrophages were cultivated on coverslips in 24-well tissue culture plates for 24 h. Cells were infected with MTB-GFP and treated with MC3465. After 4 h and 48 h, cells were fixed with 4% paraformaldehyde for 1 h at RT. LC3 labeling was performed as previously described [63]. LC3B puncta were analyzed using a Leica TCS SP8 confocal microscope and quantified using ImageJ. Dot plots represent the mean values of at least 100 cells. Error bars depict the standard deviation.

### Western blot analysis

Cells were lysed with RIPA buffer (Thermo Fisher) containing protease inhibitor cocktails (Roche) and stored at −80°C. Protein concentration was determined using the BCA protein assay kit (Thermo Fisher) following the manufacturer's instructions. About 10 μg of protein was loaded onto a NUPAGE 4% to 12% Bis-Tris polyacrylamide gel (Thermo Fisher) and transferred to PVDF membranes (iBlot 2, Thermo Fisher). The membranes were blocked with TBS-0.1% Tween20, 5% nonfat dry milk for 1 h at RT and then incubated overnight with primary antibodies against α-β-Tubulin (Cell Signaling) and LC3 (Abcam). Membranes were washed in TBS-Tween and incubated with secondary HRP-conjugated antibody (GE Healthcare) at RT for 1 h. Membranes were washed and incubated with ECL reagents (Thermo Fisher). Band detection was performed using the iBright CL1500 Imaging System (Thermo Fisher).

## RNA isolation, library preparation, and sequencing

Total RNA from macrophages was extracted using QIAzol lysis reagent (Qiagen) and purified over RNeasy columns (Qiagen). The quality of all samples was assessed with an Agilent 2100 bioanalyzer (Agilent Technologies) to verify RNA integrity. Only samples with good RNA yield and no RNA degradation (ratio of 28S to 18S, >1.7; RNA integrity number, >9) were used for further experiments. cDNA libraries were prepared with the Illumina TruSeq Stranded mRNA and the IDT for Illumina TruSeq UD Indexes and were sequenced on an Illumina NovaSeq 6000 systems at the CHU Sainte-Justine Integrated Centre for Pediatric Clinical Genomics (Montreal, Canada). Raw reads derived from the sequencing instrument are clipped for adapter sequence, trimmed for minimum quality (Q30) in 3′ and filtered for minimum length of 32 bp using Trimmomatic [72]. Surviving read pairs were aligned to GRCh38 by the ultrafast universal RNA-seq aligner STAR [73] using the recommended 2 passes approach. Aligned RNA-Seq reads were assembled into transcripts, and their relative abundance was estimated using Stringtie [74]. All of the above processing steps were accomplished through the GenPipes framework [75]. Exploratory analysis was conducted using various functions and packages from R and the Bioconductor project [76]. Differential expression was conducted using both edgeR [77] and Limma [78].

## Intracellular zinc and copper labelling

Cells were fixed and stained with the cell permeant fluorescent FluoZin 3-AM (Thermo Fisher; 2 μg/mL) for 50 min in PBS at RT. Cu+ was labeled using the fluorescent BioTracker Green copper dye (Sigma-Aldrich, 5 μM) for 3 h at 37°C. Subsequently, cells were washed and fixed before analysis. Images were acquired with the automated fluorescence microscope Opera Phenix High-Content Screening System, with a 40×/NA 0.6 water objective. Fluorescence intensity was analyzed in Columbus Conductor or Signals Image Artist software (PerkinElmer) on at least 400 different cells.

## ROS assay

Cells were incubated with MitoSOX Red (5 μM, Thermo Fisher) for 10 min at 37°C or with CellROX (5 μM, Thermo Fisher) for 30 min at 37°C. The cells were then washed with PBS and fixed. Images were captured using an automated fluorescence microscope, Opera Phenix High-Content Screening System, with a 40×/NA 0.6 water objective. Fluorescence intensity was analyzed using Signals Image Artist software (PerkinElmer).

## Quantitative reverse transcription PCR (RT-qPCR)

Reverse transcription of mRNA to cDNA was done using SuperScript III Reverse Transcriptase (Thermo Fisher) followed by amplification of cDNA using Power SYBR Green PCR Master Mix (Thermo Fisher). Reactions were performed using a StepOnePlus Real-Time PCR System Thermal Cycling block (Applied Biosystems). The relative gene expression levels were assessed according to the $2^{-\Delta Ct}$ method [79].

The primers used in this study are *ctpC* (*ctpC*-F: TCACCATTTTCACCGGGTAT; *ctpC*-R: GATGTTGAGCAACCACAGGA), *ftsZ* (ftsZ-F: CGGTATCGCTGATGGATGCTTT;ftsZ-R: CGGACATGATGCCCTTGACG), *gapdh* (*gapdh*-F: AATGAAGGGGTCATTGATGG; *gapdh*-R: AAGGTGAAGGTCGGAGTCAA), and *sirt2* (*sirt2*-F: TCACACTGCGTCAGCGC-CAG; *sirt2*-R: GGGCTGCACCTGCAAGGAG).

## Bliss independence model

Bliss independence model was used to calculate the excess over Bliss model for growth inhibition [38]. The model's formula is expressed as:

$$\hat{y}ab = ya + yb - yayb$$

where $\hat{y}ab$ is the Bliss-predicted response when drugs A and B are combined; $ya$ is the observed response to drug A alone; and $yb$ is the observed response to drug B alone.

The Excess over Bliss score ($I$) was then calculated as $I = yab - \hat{y}ab$.

The sign of $I$ indicates the nature of the drug interaction:

- $I = 0$: The drugs act independently.

- $I > 0$: Synergistic interaction (greater-than-expected effect).

- $I < 0$: Antagonistic interaction (less-than-expected effect).

## In vivo administration of MC3465 and MC3466

Seven-week-old female C57BL/6J mice were purchased from Charles River Laboratories and maintained according to Institut Pasteur guidelines for laboratory animal husbandry.

Compounds were solubilized daily in 0.5% DMSO + 4.5% Cremophor EL + 90%NaCl. Mice were treated with MC3465 and MC3466 by intraperitoneal injections for 2 wk, 6 d per weeks. Mice were deeply anesthetized with a cocktail of ketamine (Merial) and xylasine (Bayer) before cardiac puncture. Blood was collected in heparin-coated tubes. AST and ALT were quantified using the RefloverPlus analyser.

Mice were infected via aerosol, generated from a suspension containing $5 \times 10^6$ CFUs/ml of H37Rv to obtain an expected inhaled dose of 100 to 200 bacilli/lungs. One day after infection, 2 mice were killed, and the numbers of CFU were determined. After 7 d, mice were treated with MC3466 by intraperitoneal injections and/or with rifampicin (10 mg/kg) by gavage for 2 wk, 6 d per weeks. CFUs in lungs of infected animals were determined after 21 d postinfection.

## Histopathology

Lung samples were fixed in formalin for 7 d and embedded in paraffin. Paraffin sections (4-μm thick) were stained with hematoxylin and eosin (HE). Slides were scanned using the AxioScan Z1 (Zeiss) system, and images were analyzed with the Zen 2.6 software. Histopathological lesions were qualitatively described and when possible scored, using (i) distribution qualifiers (i.e., focal, multifocal, locally extensive, or diffuse); and (ii) a 5-scale severity grade, i.e., 1: minimal, 2: mild, 3: moderate, 4: marked, and 5: severe. The analysis was conducted by a trained pathologist from the histopathology core facility at the Institut Pasteur, who was blinded to the treatment group status of individual animals. For the histological heatmaps, the scores were determined as follows: the percentage of abnormal zone was estimated from low magnification images of scanned slides. All other scores were established at higher magnification (20 to 40× in the Zen program); the interstitial and alveolar syndrome scores reflected the extent of the syndrome, while the inflammation seriousness represented an evaluation of the intensity of the inflammatory reaction, i.e., abundance of inflammatory cells and exudate, conservation, or disruption of the lung architecture; the bronchiolar epithelium alteration score was derived from both the extent and the severity of the lesions.

### Enzyme-linked immunosorbent assay, ELISA

The concentration of IL-1β in lung lysates was determined in duplicate by specific sandwich ELISA as described by the manufacturer (R&D Systems).

### Quantification and statistical analysis

Data were expressed as mean ± standard deviation (shown as error bar). Statistical analyses were performed with Prism software (GraphPad Software), using the *t* test and one-way analysis of variance (ANOVA) as indicated in the figure legends. Differences between groups were examined statistically as indicated (*$p < 0.05$, **$p < 0.01$, ***$p < 0.001$, and ****$p < 0.0001$). Results were considered statistically significant with a *p*-value $< 0.05$.

## Supporting information

**S1 Table. Epigenetics compound library used in this study.**
(DOCX)

**S2 Table. List of differentially expressed genes by uninfected macrophages after 4 h of MC3465 treatment.**
(XLSX)

**S3 Table. List of differentially expressed genes by MTB-infected macrophages after 4 h of MC3465 treatment.**
(XLSX)

**S4 Table. List of differentially expressed genes by uninfected macrophages after 24 h of MC3465 treatment.**
(XLSX)

**S5 Table. List of differentially expressed genes by MTB-infected macrophages after 24 h of MC3465 treatment.**
(XLSX)

**S6 Table. List and structure of MC3465 analogs.**
(PDF)

**S1 Fig. Experimental validation of the quality of the high-throughput screening. (A)** Human macrophages were cultured in a 384-well plate and treated with different concentrations of DMSO. After 96-h treatment, cells were labeled with Hoechst 33342 and HCS Cell-Mask Blue. Fluorescence was analyzed by the Opera Phenix Plus High-Content Screening System. Enumeration of cells per well was performed using Columbus image analysis software. **(B)** MTB was grown in culture liquid medium in the presence of various concentration of DMSO. The number of intracellular bacteria was enumerated at 96 h posttreatment. **(C)** MTB-infected macrophages in a 384-well plate were treated with DMSO or RIF. After 96 h of treatment, cells were labeled as in (A). Fluorescence was analyzed by the Opera Phenix Plus High-Content Screening System. Quantification of GFP staining and enumeration of cells were performed using Columbus image analysis software. Each dot represents a well. **(D, E)** Macrophages were infected with MTB at various MOI. The number of intracellular bacteria was enumerated at 96 h postinfection by **(D)** counting the number of CFU or **(E)** calculating the GFP area as determined in (C). **(F)** MTB-infected macrophages in 384-well plate were treated with DMSO, MC3465 (10 μM), or RIF (1 μg/ml). After 96 h treatment, cells were labeled as in (A). Each dot represents a well. The z-score was calculated. Panels A-F: One representative experiment (of at least 2) is shown. *t* Test was used. Error bars represent the

mean ± SD. ****$p < 0.0001$. The data underlying the graphs shown in the figure can be found in S7 Data.
(EPS)

**S2 Fig. MC3465 was nontoxic and limited intracellular growth of MTB, independently of macrophage polarization. (A)** Naive macrophages were treated with different concentrations of MC3465. After 96 h, cell viability was assessed using the MTT assay according to the manufacturer's instructions. **(B)** Human monocytes were differentiated with GM-CSF or M-CSF for 6 d. Cells were then infected with MTB and treated with MC3465 (10 μM) for an additional 4 d. Cells were lysed and bacteria were enumerated by CFU. **(C)** Macrophages were infected with *E. coli* (strain HB101). After 1 h, cells were washed in PBS, and gentamycin (50 μg/ml) was added to kill extracellular bacteria. After an additional 1 h, cells were washed and incubated in fresh complete medium containing gentamycin (5 μg/ml) with DMSO or MC3465 (10 μM). Intracellular bacteria were enumerated at 0, 4, and 24 h postinfection. **(D)** MTB was grown in culture liquid medium in the presence of various concentration of MC3465. The number of intracellular bacteria was enumerated at 96 h posttreatment through CFU counting. Panels A-D: One representative experiment (out of 2) is shown. Error bars represent the mean ± SD. *t* Test was used. ** $p < 0.01$, *** $p < 0.001$. The data underlying the graphs shown in the figure can be found in S8 Data.
(EPS)

**S3 Fig. MC3465 constrained the intracellular growth of MTB independently of SIRT2. (A)** MTB-infected macrophages were treated with different concentrations of MC3465 and of 2 SIRT2 inhibitors, namely, AGK2 and SirReal2. After 96 h, cells were lysed and bacteria were enumerated by CFU. **(B)** Macrophages were preincubated with AGK2 (10 μM) and MC3465 (10 μM). After 2 h, the cells were infected with MTB in the continuous presence of AGK2 and MC3465. The enumeration of intracellular bacteria was performed 96 h after infection. **(C, D)** *SIRT2* expression was down-regulated in macrophages using siRNA-mediated gene silencing. **(C)** Relative *SIRT2* expression measured by RT-qPCR in SIRT2-silenced cells. Relative expression levels were normalized to the *GADPH* gene. Scramble siRNA (scRNA) was used as a negative control. **(D)** WT and SIRT2-silenced macrophages were infected with MTB and treated with MC3465 (10 μM). After 96 h, the number of intracellular bacteria was counted. **(E)** WT MEFs or SIRT2$^{-/-}$ MEFs were infected with MTB and treated with MC3465 (10 μM). After 48 h, cells were lysed and the number of intracellular bacteria was enumerated. Panels A-E: One representative experiment (of 2) is shown. One-way ANOVA test (S3A and S3B) and *t* test (S3E) were used. Error bars represent the mean ± SD. ns, not significant, **$p < 0.01$, ***$p < 0.0001$. The data underlying the graphs shown in the figure can be found in S9 Data.
(EPS)

**S4 Fig. The MC3465 treatment did not reduce intracellular MTB growth through the inhibition of the mycobacterial SIRT2-like protein, Rv1151c. (A)** Rv*Δ1151c* knock-out strain was obtained by performing DNA recombination. Three different couples of primers were used: 5′-actacagctggatggattccg-3′ and 5″-atttaaatcgctcagtggaacg -3′ (amplification for if WT H37Rv: 0 bp, amplification if H37RvΔ*1151c*: 1531 bp), 5′- atttaaataacgaaaggctcagtc-3′ and 5′-acacgccagcgtcagcaatc-3′ (amplification for if WT H37Rv: 0 bp, amplification if H37RvΔ*1151c*: 1857 bp), 5′-actacagctggatggattccg-3′ and 5′-acacgccagcgtcagcaatc-3′ (amplification for if WT H37Rv: 2616 bp, amplification if H37RvΔ*1151c*: 2550 bp). **(B)** Growth of MTB H37Rv or RvΔ1151c in liquid culture medium, determined by OD$_{600}$. **(C)** Human monocyte-derived macrophages were infected with either MTB WT (H37Rv) or a Rv1151c null mutant (RvΔ*1151c*). Cells were then treated with MC3465 and MC3466 (10 μM). After 0 or 96 h,

bacteria were enumerated by CFU. One representative experiment (of 2) is shown. *t* Test was used. Error bars represent the mean ± SD. ****$p < 0.0001$. The data underlying the graphs shown in the figure can be found in S10 Data.
(EPS)

**S5 Fig. MC3465 did not induce autophagy in MTB-infected macrophages. (A)** Detection by indirect immunofluorescence of LC3 (red) in MTB (green) infected macrophages, treated with MC3465 (10 μM) for 4 h (scale bar: 10 μm). DAPI (blue) was used to visualize nuclei. **(B)** Determination of the number of LC3-positive puncta per cell (unpaired *t* test) after 4 h and 48 h treatment with MC3465 (10 μM). **(C)** Western blot analysis of LC3 and α/ß-tubulin in MTB-infected cells treated with MC3465 (10 μM) and bafilomycin A1 (BAF, 100 nM) for 4 and 48 h. **(D)** MTB-infected macrophages were left untreated or incubated with MC3465 (10 μM) plus BAF (100 nM), 3-MA (5 mM), or CQ (40 μM). After 4 d, the number of intracellular bacteria was enumerated. Panels A-D: One representative experiment (of at least 2) is shown. Error bars represent the mean ± SD. *t* Test was used. ns, not significant, *** $p < 0.001$. The data underlying the graphs shown in the figure can be found in S11 Data.
(EPS)

**S6 Fig. MC3465 treatment did not increase intracellular reactive oxygen species (ROS) levels.** MTB-infected macrophages were cultured in a 384-well plate and treated with DMSO or MC3465 (10 μM). Total and mitochondrial ROS staining were performed after 4 and 24 h using respectively CellROX deep red (5 μM) **(A, B)** or MitoSOX (5 μM) **(C, D)**. Fluorescence was analyzed by the Opera Phenix Plus High-Content Screening System. Enumeration of cells per well was performed using Columbus image analysis software. Scale bar, 20 μm. **(E)** MTB-infected cells were treated with MC3465 with either the antioxidant GSH (1 mM) or with its precursor NAC (10 mM). The number of intracellular bacteria was enumerated at 96 h post-treatment. Panels A-E: Data are representative of 2 independent experiments. Error bars represent the mean ± SD. The data underlying the graphs shown in the figure can be found in S12 Data.
(EPS)

**S7 Fig. Gene ontology enrichment analysis of genes down-regulated by MC3465 treatment.** Gene ontology enrichment analysis of genes whose expression is down-regulated by MC3465 treatment in MTB-infected cells, using the Cytoscape app ClueGO (p.value<0,05; LogFC<-0,5 or LogFC>0,5) after 4 h **(A)** and 24 h **(B)**.
(EPS)

**S8 Fig. Free zinc is released upon MC3465 treatment.** Naive macrophages were incubated with ZNSO$_4$ (50 and 100 μM), MC3465 (10 and 50 μM), or the zinc-chelating agent TPEN (2.5 μM) during 4 h. **(A)** Cells were fixed and stained with FluoZin 3-AM. Scale bar, 20 μm. **(B)** Average FluoZin 3-AM signal intensity, for at least 400 cells per condition. **(C, D)** MTB-infected macrophages were treated with MC3465 (10 μM) during 24 h and 96 h. Cells were then fixed and stained with FluoZin 3-AM. **(C)** Representative images of FluoZin 3-AM staining. Scale bar, 20 μm. **(D)** Average FluoZin 3-AM signal intensity for at least 400 cells per condition. Panels A-D: Data are representative of 3 independent experiments. One-way ANOVA test was used. Error bars represent the mean ± SD. **$p < 0.01$, ****$p < 0.0001$. The data underlying the graphs shown in the figure can be found in S13 Data.
(EPS)

**S9 Fig. The treatment with MC3465 did not induce a burst of copper. (A)** MTB-infected macrophages were treated with MC3465 (10 μM) for 4 and 24 h, and copper ions (Cu+) were

subsequently stained using the fluorescent probe BioTracker Green Copper dye (5 μM) as described by the manufacturer. Scale bar, 10 μm. **(B)** Average signal intensity of BioTracker Green Copper Dye, calculated for at least 400 cells per condition. Data are representative of 2 independent experiments. One-way ANOVA test was used. Error bars represent the mean ± SD. ns, not significant. The data underlying the graphs shown in the figure can be found in S14 Data.
(EPS)

**S10 Fig. Treatment with inactive MC3465 analogs does not result in free zinc released. (A)** MTB was grown in culture liquid medium in the presence of MC3466 (10 and 50 μM). The number of intracellular bacteria was enumerated at 96 h posttreatment through CFU counting. **(B)** Structures of inactive MC3465 analogs. **(C)** Macrophages were infected with GFP expressing MTB and incubated with inactive MC3465 analogs identified in Fig 5A (10 μM). After 96-h treatment, cells were labeled with Hoechst 33342 and HCS CellMask Blue. Fluorescence was analyzed by the Opera Phenix Plus High-Content Screening System. Quantification of GFP staining was performed using Columbus image analysis software. The areas of intracellular bacteria ($px^2$: $pixels^2$) were expressed as the percentage in compound-treated cells compared to cells incubated with DMSO. **(D)** Representative image of FluoZin 3-AM staining. MTB-infected macrophages were incubated with MC3465 analogs (10 μM) during 24 h. Cells were then fixed and stained with FluoZin 3-AM. **(E)** Average FluoZin 3-AM signal intensity for at least 400 cells per condition. Panels C-E: One representative experiment (of 2) is shown. Error bars represent the mean ± SD. The data underlying the graphs shown in the figure can be found in S15 Data.
(EPS)

**S11 Fig. Susceptibility and resistance of MTB strains to MC3465, MC3466, RIF, and BDQ.** **(A-C)** WT MTB H37Rv was incubated in liquid culture medium with different concentrations of MC3465 or MC3466 **(A)**, RIF **(B),** or BDQ **(C)** alone or in association with MC3465 or MC3466 (10 μM). The growth was determined by $OD_{600}$. One representative experiment (of at least 3) is shown. Error bars represent the mean ± SD. The data underlying the graphs shown in the figure can be found in S16 Data.
(EPS)

**S12 Fig. MC3465 and MC3466 were well tolerated in mice. (A)** The murine macrophage cell line, RAW 264.7, was infected with MTB and treated with MC3465 or MC3466 (10 μM). After 0 or 48 h, bacteria were enumerated by CFU. **(B-D)** Eight-week-old female C57BL/6J mice received different concentrations of MC3465 and MC3466 by intraperitoneal injection 6 d per week during 2 wk. **(B)** Weight was evaluated before each injection. **(C, D)** Blood was drawn before mouse killing. **(C)** ALT and **(D)** AST levels were measured according to the manufacturer's instructions. Error bars represent the mean ± SD. One-way ANOVA test was used. ns, not significant, ***$p < 0.001$, ****$p < 0.0001$. The data underlying the graphs shown in the figure can be found in S17 Data.
(EPS)

**S1 Data. Supporting data files for Fig 1.** These files contain the numerical values used to generate the graphs in Fig 1, the statistical analyses, and the $p$-value.
(XLSX)

**S2 Data. Supporting data files for Fig 3.** These files contain the numerical values used to generate the graphs in Fig 3, the statistical analyses, and the $p$-value.
(XLSX)

**S3 Data. Supporting data files for Fig 4.** These files contain the numerical values used to generate the graphs in Fig 4, the statistical analyses, and the *p*-value.
(XLSX)

**S4 Data. Supporting data files for Fig 5.** These files contain the numerical values used to generate the graphs in Fig 5, the statistical analyses, and the *p*-value.
(XLSX)

**S5 Data. Supporting data files for Fig 6.** These files contain the numerical values used to generate the graphs in Fig 6, the statistical analyses, and the *p*-value.
(XLSX)

**S6 Data. Supporting data files for Fig 7.** These files contain the numerical values used to generate the graphs in Fig 7, the statistical analyses, and the *p*-value.
(XLSX)

**S7 Data. Supporting data files for S1 Fig.** These files contain the numerical values used to generate the graphs in S1 Fig, the statistical analyses, and the *p*-value.
(XLSX)

**S8 Data. Supporting data files for S2 Fig.** These files contain the numerical values used to generate the graphs in S2 Fig, the statistical analyses, and the *p*-value.
(XLSX)

**S9 Data. Supporting data files for S3 Fig.** These files contain the numerical values used to generate the graphs in S3 Fig, the statistical analyses, and the *p*-value.
(XLSX)

**S10 Data. Supporting data files for S4 Fig.** These files contain the numerical values used to generate the graphs in S4 Fig, the statistical analyses, and the *p*-value.
(XLSX)

**S11 Data. Supporting data files for S5 Fig.** These files contain the numerical values used to generate the graphs in S5 Fig, the statistical analyses, and the *p*-value.
(XLSX)

**S12 Data. Supporting data files for S6 Fig.** These files contain the numerical values used to generate the graphs in S6 Fig, the statistical analyses, and the *p*-value.
(XLSX)

**S13 Data. Supporting data files for S8 Fig.** These files contain the numerical values used to generate the graphs in S8 Fig, the statistical analyses, and the *p*-value.
(XLSX)

**S14 Data. Supporting data files for S9 Fig.** These files contain the numerical values used to generate the graphs in S9 Fig, the statistical analyses, and the *p*-value.
(XLSX)

**S15 Data. Supporting data files for S10 Fig.** These files contain the numerical values used to generate the graphs in S10 Fig, the statistical analyses, and the *p*-value.
(XLSX)

**S16 Data. Supporting data files for S11 Fig.** These files contain the numerical values used to generate the graphs in S11 Fig, the statistical analyses, and the *p*-value.
(XLSX)

**S17 Data. Supporting data files for S12 Fig.** These files contain the numerical values used to generate the graphs in S12 Fig, the statistical analyses, and the *p*-value.
(XLSX)

**S1 Raw Images. Raw images.**
(PDF)

## Acknowledgments

We thank Charles Privé (CHU Sainte-Justine Integrated Centre for Pediatric Clinical Genomics, Montreal, Canada) for his technical support.

## Author Contributions

**Conceptualization:** Alexandra Maure, Claude Gutierrez, Olivier Neyrolles, Ludovic Tailleux.

**Formal analysis:** Alexandra Maure, Emeline Lawarée, Saideep Gona, Alexandre Giraud-Gatineau, Anne Danckaert, David Hardy, Ludovic Tailleux.

**Funding acquisition:** Nathalie Aulner, Antonello Mai, Roland Brosch, Dante Rotili, Ludovic Tailleux.

**Investigation:** Alexandra Maure, Emeline Lawarée, Francesco Fiorentino, Alexandre Pawlik, Matthew J. G. Eldridge, Wafa Frigui, Mélanie Hamon, Ludovic Tailleux.

**Methodology:** Alexandra Maure, Emeline Lawarée, Claude Gutierrez, Olivier Neyrolles.

**Resources:** Francesco Fiorentino, Antonello Mai, Dante Rotili.

**Supervision:** Nathalie Aulner, Antonello Mai, Mélanie Hamon, Luis B. Barreiro, Priscille Brodin, Roland Brosch, Dante Rotili, Ludovic Tailleux.

**Validation:** Alexandra Maure, Camille Keck.

**Visualization:** Alexandra Maure, Ludovic Tailleux.

**Writing – original draft:** Ludovic Tailleux.

**Writing – review & editing:** Alexandra Maure, Emeline Lawarée, Ludovic Tailleux.

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
