## [Editor Report · Decision Letter 0]

19 Jul 2023

Dear Dr Tailleux, 

Thank you for submitting your manuscript entitled "An oxadiazole-based compound potentiates anti- tuberculosis treatment by increasing host resistance via zinc poisoning" for consideration as a Research Article by PLOS Biology.

Your manuscript has now been evaluated by the PLOS Biology editorial staff, as well as by an academic editor with relevant expertise, and I am writing to let you know that we would like to send your submission out for external peer review.

Once your full submission is complete, your paper will undergo a series of checks in preparation for peer review. After your manuscript has passed the checks it will be sent out for review. To provide the metadata for your submission, please Login to Editorial Manager (https://www.editorialmanager.com/pbiology) within two working days, i.e. by Jul 21 2023 11:59PM.

Kind regards,

Paula

---

Senior Editor

PLOS Biology

---

## [Decision Letter · Decision Letter 1]

5 Sep 2023

Dear Dr. Tailleux,

Thank you for your patience while your manuscript "An oxadiazole-based compound potentiates anti- tuberculosis treatment by increasing host resistance via zinc poisoning" was peer-reviewed at PLOS Biology. It has now been evaluated by the PLOS Biology editors, an Academic Editor with relevant expertise, and by several independent reviewers. 

In light of the reviews, which you will find at the end of this email, we would like to invite you to revise the work to thoroughly address the reviewers' reports.

As you will see below, the reviewers find your work interesting, but they all find issues that would need to be solved before further consideration at PLOS Biology. In particular, we think it is important that you address the causation between the compound and zinc intoxication with additional experiments.

Given the extent of revision needed, we cannot make a decision about publication until we have seen the revised manuscript and your response to the reviewers' comments. Your revised manuscript is likely to be sent for further evaluation by all or a subset of the reviewers.

**IMPORTANT - SUBMITTING YOUR REVISION**

*Re-submission Checklist*

*Published Peer Review*

*PLOS Data Policy*

*Blot and Gel Data Policy*

Sincerely,

Paula

---

Senior Editor

PLOS Biology

REVIEWS:

Reviewer #1: Tuberculosis and antimicrobial resistance

Reviewer #2: Mouse and rabbit models of tuberculosis

Reviewer #1: The manuscript by Maure et al. reports the identification of a novel 1,2,4-oxadiazole-based compound that restricts the growth of Mycobacterium tuberculosis in macrophages, likely through zinc intoxication. In combination with known TB drugs, this compound reduces MTB burden when administered in mice in the acute phase of infection. The manuscript is cogent, the results presented are overall solid and the conclusions sound. However, some of the points below must be addressed while others—although there shouldn't be necessarily addressed—may strengthen the overall robustness of the data presented in the manuscript.

Major points that should be addressed/discussed:

- Figure 1: To demonstrate that MC3465 might have specificity towards MTB, the authors show that MC3465 is inactive against Salmonella and Listeria. As accumulation of intracellular concentration of zinc has been shown to be bactericidal against E. coli (Ref. 27), why has the compound not been tested on E. coli?

- Figure 3: Why do similar intracellular concentrations of zinc (panel B: - TPEN - MC3465 and + TPEN + MC3465) lead to such differences in CFU counts (panel F)? 

- Figure 4: To add to the lack of correlation between zinc intracellular concentrations and the inhibitors' efficacy, MC3465 is static while MC3466 is cidal, although the former leads to higher accumulation of intracellular zinc than the latter; if this is due to experimental variability, both compounds should be tested in the in same experiment for their effect on the level of intracellular zinc and CFU counts.

- As MC3465 must have been used as a reference, please indicate where MC3465 is in panel 4A.

- Figure 6: Does the compound have an effect when administered during the chronic phase of infection? Is the treatment intended for prophylaxis? The rationale of the design of the experiment should be unambiguously stated. In Panel B, the addition of arrows would help guide the naïve eye. 

- Page 16: "improves the healing of lesions". This is an overstatement; the authors did not show that there were any lesions in the first place.

Minor points that should be addressed:

- "Epigenetic compounds" -reformulate. 

- In page 5, in the sentence that says: "A 25% and 55% decrease of CFUs", are these numbers correct? For clarity, all variations should be expressed in n fold-reduction.

- Figure 4: MC3465 seems to have less of an effect as compared to panel 1C—is it because the DMSO control grew less? If so, this should be mentioned. 

Non-essential, additional work that would strengthen the manuscript:

- In figure 4, including an inactive analog as a negative control would have been appropriate.

- Does the CtpC mutant that cannot efflux zinc have increased sensitivity to MC3465 in macrophage?

- Would the authors anticipate that mutants with mutations in ctpC that leads to overexpression of CtpC could be a mechanism of resistance to MC3465? Are there any such mutants found in clinical isolates—that evolved to counter host immunity—that could be tested for their sensitivity to MC3465?

Reviewer #2: In the manuscript, the authors have screened a library of 157 epigenetic-modifying molecules to identify novel drugs that restrict Mtb growth in macrophages. The authors propose that the top hit molecules MC3465 and MC3466 are not inhibitors of the SIRT2 histone deacetylase, but rather they induce Zn accumulation inside macrophages to inhibit Mtb replication. Additionally, MC3466 was shown to potentiate the lethality of rifampicin inside macrophages and in an acute mouse model of TB disease. 

Major Concerns:

1. While the study shows that MC3465 and MC3466 lead to Zn accumulation in cells which is inhibitory to Mtb, the study falls short of identifying a mechanism for the drug-mediated increase intracellular Zn levels and decrease in metallothionein levels. It would be valuable to know if MC3465 modulates the intracellular levels of other metals such as Cu and Fe? In Mtb -infected macrophages, the Fluozin-3 signal was inhibited by apocynin, an inhibitor of the NADPH phagocyte oxidase, strongly suggesting that the oxidative burst observed during mycobacterial infection leads to the dissociation of zinc from intracellular Zn2+-MT complexes. Does MC3465 induce phox expression and/or its activity?

2. Fig. 1a. High throughput screen. A z-value should be provided to measure the robustness of the HTS assay shown in Figure 1A (PMID: 10838414). Also, the data on positive control and negative controls used in the screen should be included. Additionally, in this imaging-based assay, controls where the GFP area/cell are correlated with CFU and MOI would be important to include.

3. Fig. 1a. Human MDMs were used. Were these donors tested for latent TB infection (LTBI) by TST and/or IGRA? The authors should mention this in the methods.

4. P. 5, first full paragraph. It is not clear how molecule-induced cytotoxicity was determined by Hoechst 33342 staining. This dye is cell-permeant and stains both live and dead cells. 

5. Fig. 1C, 4C: Mtb CFU increase in the presence of MC3465. But the CFU increase is slower than it is with no drug. The authors need to state this clearly. Also, on p. 11 lines 8-13, how can one calculate an IC50 when the bacillary count is actually increasing? Presumably, this was done by setting up proportions to the CFU counts of untreated controls at each time point. In any event, the authors need to state clearly how IC50 was determined.

6. The authors should demonstrate that MC3465 does not directly inhibit SIRT2 activity in vitro as shown in Ref. 18 to rule out this possibility. The previous study (ref. 18) also shows that AGK2 decreases Mtb CFU burden; however, this killing activity was absent in Figure S2A. The authors should address this apparent discrepancy.

7. Figure 2A. The authors state there are 4 groups: DMSO-naive, DMSO-infected, MC3465-naive, MC3465 infected; however, it is not clear what the comparator groups are in Fig. 2a. Please clarify this. Additionally, the cutoff should be at least >1.5 absolute-fold change (log2 FC>0.6) and FDR <0.05, not p-value, so these data should be re-analyzed with appropriate cutoff values. Also, the list of DEGs is not present as a supplementary file, and this list should be added. 

8. The authors did not mention why downregulated genes were selected for analysis and what the upregulated genes in the RNA-Seq analysis were. On p. 8, second to last sentence, the term "significantly enriched" is used; the p-values associated with this statement should be provided. Figures 2b and 2c are cytoscape images that do not add meaningful information; these could easily be moved to the supplement

9. Page 9, paragraph 1. Ref 27 shows elevated metallothionein (MT) gene expression during Mtb infection is associated with high Zn levels in host cells. Thus, the literature suggests that MT genes are induced at higher concentrations of Zn. The authors therefore need to clarify their claim that lower metallothionein gene expression leads to high labile Zn. Additionally, high concentrations of Zn induce many epigenetic changes and numerous signaling mechanisms in human cells that can indirectly affect host-bacteria interaction and this confounder should at least be discussed (PMID: 26959009)

10. Figure 3. While Zn is highly accumulated at 4 h, the CFU analysis is done at 96 h. Zn accumulation and CFU should be done at the same timepoints. The authors should provide the Zn levels at later time points (96 h) where there is a bacteriostatic effect to provide direct proof of Zn intoxication.

11. Figure 3c and 3d. Are the results shown also observed at 96 h? How exactly the phagosomes were counted/determined? 

12. Figures 3E and 3F. The concentrations of ZnSo4 and MC3465 needs to be stated. The upregulation of ctpC is marginal and it is unclear if it is statistically significant. Multiple timepoints need to be tested for this experiment. In Ref. 27, the addition of 0.5 mM Zn led to very high ctpC expression, and the authors should offer explanations for the very modest induction reported here. Additionally, either 16S or sigA gene expression is usually used for normalization, rather than the blank which in this case is DMSO. 

13. Fig. 6. For mouse infection experiments, the day 0 CFU data would be helpful. Additionally the authors should address the 1000-fold variation in bacterial burden after MC3466-only treatment. For Fig. 6c and p. 23, the authors should clarify who performed the histopathological analysis (was it a trained pathologist?) and whether or is not the scorer was blinded to the identity of the samples.

14. RNA seq data show reduced expression of genes involved in TNF, type I IFN signaling, and innate immunity alterations. This can reduce inflammation and tissue damage inside the host leading to better treatment outcomes during Mtb infection. Authors should check levels of these cytokines in lungs and also, Zn levels inside mice to support their in vitro data of Zn toxicity to Mtb and disprove other possible reasons. 

15. Statistical analysis and scientific rigor. Please show the number of biological and technical replicates; this information is missing in many experiments. For Fig 1 and Fig. 3 it is impossible to tell which panel used T-test and which used ANOVA The data figures would be strengthened by showing all data points in addition to the SD . In Fig. 4c, 5a, and 5b it is unclear which groups are being compared for the p values shown. 

Minor Concerns:

16. P. 3, lines 15-16. MDR-TB is now readily treatable with the BPaL regimen. Please see Conradie et al NEJM 2022 PMID 36053506

17. Fig. S3b and P. 8, line 4: The figure shows that LC3B puncta are lower in MC3465-treated cells at the early time point (p< 0.001), while the text states that there was a minimal difference. Also, there is a discrepancy between the figure (which says the early time point is 4 h) and the text (which says that the early time point is 24 h). Additionally, a Western blot must be done to see unchanged LC3 I and II levels to detect alterations in autophagy at 4h and 96 h.

18. P. 12 line 15-17. Neither the figure nor the text state what the Bliss scores are. Please provide the actual score. The methodology should be properly written so that others can repeat the experiment if required. The color code does not match or is not clear to me.

19. P. 13, lines 5-8. It is noted that there were no adverse effects on AST and ALT, but as a dru

---

## [Editor Report · Decision Letter 2]

27 Feb 2024

Dear Dr Tailleux,

Thank you for your patience while we considered your revised manuscript "An oxadiazole-based compound potentiates anti- tuberculosis treatment by increasing host resistance via zinc poisoning" for publication as a Research Article at PLOS Biology. This revised version of your manuscript has been evaluated by the PLOS Biology editors and the Academic Editor.

Based on our Academic Editor's assessment of your revision, I am pleased to say that we are likely to accept this manuscript for publication, provided you satisfactorily address the remaining comments raised by the Academic Editor, pasted below my signature (labelled 'Comments from the Academic Editor'). In addition, please also make sure to address the following data and other policy-related requests that I have provided below (A-G):

(A) We would like to suggest the following modification to the title: 

“A host-directed oxadiazole compound potentiates anti-tuberculosis treatment via zinc poisoning in human macrophages and in a mouse model of infection"

(B) In the human ethics statement in the Methods section, please provide the specific approval number provided by the local ethics committee that reviewed and approved your study. In addition, please clarify that the participants provided a written informed consent. 

(C) You may be aware of the PLOS Data Policy, which requires that all data be made available without restriction: http://journals.plos.org/plosbiology/s/data-availability. For more information, please also see this editorial: http://dx.doi.org/10.1371/journal.pbio.1001797

-Supplementary files (e.g., excel). Please ensure that all data files are uploaded as 'Supporting Information' and are invariably referred to (in the manuscript, figure legends, and the Description field when uploading your files) using the following format verbatim: S1 Data, S2 Data, etc. Multiple panels of a single or even several figures can be included as multiple sheets in one excel file that is saved using exactly the following convention: S1_Data.xlsx (using an underscore).

-Deposition in a publicly available repository. Please also provide the accession code or a reviewer link so that we may view your data before publication. 

Figure 1A, 1C-G, 3B, 3D-E, 4A-C, 5A, 5C-F, 5H-I, 6A-D, 7A, 7C-D, S1A-F, S2A-C, S3A-E, S4B-C, S5B, S5D, S6B, S6D-E, S8B, S8D, S9B, S10A, S10C, S10E, S11A-C, S12A-D

(D) Thank you for depositing the RNA-seq in the GEO database (GSE222412). However, I note that the data is currently on hold and scheduled for release on June 1st 2024. We ask that you please make this data publicly available before publication.

(E) Please also ensure that each of the relevant figure legends in your manuscript include information on *WHERE THE UNDERLYING DATA CAN BE FOUND*, and ensure your supplemental data file/s has a legend.

(F) We require the original, uncropped and minimally adjusted images supporting all blot and gel results reported in the following Figures:

Figure S4A, S5C

We will require these files before a manuscript can be accepted so please prepare and upload them now. Please carefully read our guidelines for how to prepare and upload this data: https://journals.plos.org/plosbiology/s/figures#loc-blot-and-gel-reporting-requirements

(G) Please ensure that your Data Statement in the submission system accurately describes where your data can be found and is in final format, as it will be published as written there. 

We expect to receive your revised manuscript within two weeks. 

*Published Peer Review History*

*Press*

Sincerely,

Richard

Richard Hodge, PhD

rhodge@plos.org

COMMENTS FROM THE ACADEMIC EDITOR

After checking the revision and the ‘Response to Reviewers’, I believe the authors did a very careful job of addressing the reviewers' concerns. I have two minor comments regarding the response to reviewer 1:

"As shown in the figure below, MC3465 treatment had no effect on bacterial load. However, it is important to note that E. coli was rapidly eliminated by macrophages. At 24 hours post infection, we detected almost no bacteria. This may explain why MC3465 has no effect while E. coli is sensitive to zinc. The macrophage response with the zinc burst was probably already optimal for killing E. coli ".

"We thank the reviewer for raising this issue. Treatment with the zinc chelator TPEN rapidly and dramatically decreased FluoZin-3 AM fluorescence to near background levels, regardless of whether the cells were treated with MC3465. The cells then became highly susceptible to MTB infection. With the technique we used, we cannot precisely quantify the amount of intracellular zinc. We can speculate that zinc may be present in the phagosome at levels undetectable by FluoZin-3 AM in the absence of TPEN, and that this may in part limit MTB growth"

These arguments - even though they are somewhat speculative - should be part of the manuscript.

---

## [Editor Report · Decision Letter 3]

13 Mar 2024

Dear Ludovic,

On behalf of my colleagues and the Academic Editor, Csaba Pál, I am pleased to say that we can in principle accept your manuscript for publication, provided you address any remaining formatting and reporting issues. These will be detailed in an email you should receive within 2-3 business days from our colleagues in the journal operations team; no action is required from you until then. Please note that we will not be able to formally accept your manuscript and schedule it for publication until you have completed any requested changes.

PRESS

Kind regards, 

Richard

Richard Hodge, PhD

rhodge@plos.org

PLOS
